# Quantifying vertical wind shear effects in shallow cumulus clouds over Amazonia

Micael Amore Cecchini[1,5], Marco de Bruine[2], Jordi Vilà-Guerau de Arellano[3,4], Paulo Artaxo[6]

[1]Institute of Astronomy, Geophysics and Atmospheric Sciences, University of São Paulo, São Paulo, Brazil
[2]Institute for Marine and Atmospheric Research, Utrecht University, Utrecht, the Netherlands
[3]Meteorology and Air Quality Section, Wageningen University, Wageningen, Netherlands
[4]Atmospheric Chemistry Department, Max Planck Institute for Chemistry, Mainz, Germany
[5]Deparment of Atmospheric Science, Colorado State University, Fort Collins, United States of America
[6]Institute of Physics, University of São Paulo, São Paulo, Brazil

*Correspondence to*: Micael A. Cecchini (micael.cecchini@usp.br)

**Abstract.** This study analyses and quantifies the effects of vertical wind shear (VWS) on the properties of shallow cumulus cloud fields over Central Amazonia. We perform idealised simulations with high resolution (50 m horizontally and 20 m vertically) using the Dutch Atmospheric Large Eddy Simulation model, changing the initial conditions and large scale forcing of VWS. The resulting cloud field is analysed by applying a cloud tracking algorithm to generate Lagrangian datasets of the lifecycle of individual clouds as well as their time-varying core and margin dimensions. The reference run has no wind speed or directional shear and represents a typical day in the local dry season. Numerical experiments with moderate and high wind speed shear are simulated by adding linear increases in the wind speed of 1.2 m s$^{-1}$ km$^{-1}$ and 2.4 m s$^{-1}$ km$^{-1}$, respectively. Three additional runs are made by adding 90° of wind rotation between the surface and the top of the domain (5 km) on top of the three wind speed shear conditions. We find that clouds developing in a sheared environment have horizontal equivalent diameter increased by up to 100 m on average, but the cloud depth is reduced. Our quantification shows that VWS tends to increase the size of the cloud cores but reduces their relative area, volume, and mass fractions compared to the overall cloud dimensions. The addition of 2.4 m s$^{-1}$ km$^{-1}$ of VWS decreases the relative core area by about 0.03 (about 10% of the overall average) and its volume and mass ratios by about 0.05 (10%-25% in relative terms). Relevant for the cloud transport properties is that the updraught speed and the liquid water content are lower within the cores, and consequently so is the upward mass flux. All quantifications of mean cloud properties point to the inhibition of convective strength by VWS, therefore hampering the shallow-to-deep transition. However, open questions still remain given that the individually deepest clouds were simulated under high environmental shear, even though they occur in small numbers. This could indicate other indirect effects of VWS that have opposite effects on cloud development if found to be significant in the future.

## 1 Introduction

The Amazon rainforest is a key component of the Earth's climate system, contributing to important biogeochemical, hydrologic, and energy cycles (Silva Dias et al., 2002; Nobre et al., 2009; Artaxo et al., 2013). The widespread convection in the region represents one of the world's most important heat engines (Nobre et al., 2009), being responsible for large-scale energy and moisture transport. In terms of the South American regional climate, it is also a crucial heat source to maintain the Bolivian High, which is an important component of the South American monsoon system (Zhou and Lau, 1998; Vera et al., 2006). Under a changing global climate, the Amazon is likely to suffer a paradigm shift with significant consequences for the local and global climate (Betts et al., 2008; Lovejoy and Nobre, 2018). Understanding the cloud regimes and their sensitivity to changing environmental conditions is then crucial for future climate projections.

Shallow cumulus cloud fields have an important role in the climate of the Amazon rainforest. Similar to their deeper counterparts, shallow clouds redistribute energy and humidity both horizontally as well as vertically (Riehl et al., 1951). The evaporation at the tops of shallow clouds destabilises and moistens the atmosphere and helps break the inversion layer, contributing to the later development of deeper convection (Neggers et al., 2007; Stevens, 2007). Previous studies have pointed out the importance of low- and mid-tropospheric humidity to the shallow-to-deep convective transition (Schiro et al., 2016; Chakraborty et al., 2018; Schiro and Neelin, 2019).

Recent studies, based on aircraft and ground-based observations from the GoAmazon2014/5 ("Observations and Modeling of the Green Ocean Amazon", Martin et al., 2016) and ACRIDICON-CHUVA ("Aerosol, Cloud, Precipitation, and Radiation Interactions and Dynamics of Convective Cloud Systems"-"Cloud Processes of the Main Precipitation Systems in Brazil: A Contribution to Cloud Resolving Modelling and to the GPM (Global Precipitation Measurement)"), Wendisch et al., 2016; Machado et al., 2018) experiments, have found a unique link between the natural aerosols and clouds over the forest. It was found that high concentrations of aerosol particles in the boundary layer occur right after precipitation (Wang et al., 2016). Andreae et al. (2018) found aerosol concentrations 1 to 2 orders of magnitude higher than in the boundary layer 5-72 hours after deep convection cases. The proposed explanation is that the continuum of shallow to deep clouds transports Volatile Organic Compounds (VOC) to the Upper Troposphere (UT), where they condense and produce the high number concentrations observed. The findings in Wang et al. (2016) indicate that such particles are reintroduced into the boundary layer by precipitation-related downdraughts, where they continue to grow by the condensation of biogenic VOCs (BVOCs). On the other hand, Machado et al. (2021) argue that the downdraughts within deep convection are not at high enough altitudes and suggest that there are likely other sources of aerosol particles.

Regardless of the exact nature of the processes responsible for transporting and generating aerosol particles in the Amazon, it is very likely that deep convective clouds and also shallow clouds play an important role. Therefore, it is desirable to understand the conditions that favour the formation and growth of convective clouds over the region. While the primary mechanism for deep convection seems to be the column water vapour (CWV) amount (Zhuang et al., 2017; Chakraborty et al., 2018; Schiro and Neelin, 2019), vertical wind shear (VWS) is another environmental control of cloud growth that has yet

to be fully understood. Most often, the VWS effects are analysed in the context of deep convection or mesoscale systems (Houze, 2018), with shallow clouds being relatively underrepresented in the literature. In addition, a significant portion of VWS studies for shallow clouds is focused on marine trade wind cumulus (e.g., Yamaguchi et al., 2019; Helfer et al., 2020; Drueke et al., 2021) rather than land tropical rainforests.

Perhaps the most direct way in which VWS affects cumulus cloud growth is by tilting them vertically (Malkus, 1952; Asai, 1964). This tilting perturbs the vertical ascent of cloud parcels, usually reducing the updraught speed (e.g., Helfer et al., 2020). Recently, Drueke et al. (2021) found that VWS promotes cloud core dilution. In contrast, a monotonic increase in the background wind speed has been suggested to increase cloud depth (Nuijens and Stevens, 2012). For the Amazonian region, Chakraborty et al. (2018) have found that shallow cumulus convection is more likely to occur under high low-level (up to ~3 km height) VWS ($\gtrsim$2.0 m s$^{-1}$ km$^{-1}$). This would be consistent to the previous findings of weaker updraughts and cloud cores under sheared environments. However, Zhuang et al. (2017) found a positive correlation between low-level VWS and the occurrence of deep convection in Amazonia during the dry season. Yamaguchi et al. (2019) show that VWS can have different effects on cloud growth due to contrasting effects on evaporation. While cloud tilting leads to more evaporation of the clouds, VWS also enhances cloud clustering and thus protects them from evaporation. Such duality indicates that the effects of VWS on cumulus cloud fields can be complex and with opposite effects.

Another potential indirect effect of VWS is related to its role in the formation of the morning boundary layer (BL). Henkes et al. (2021) have analysed this issue for the Amazonian region in the context of the shallow-to-deep convective transition and found a significant change in VWS prior to the cloud formation. On days characterised by deep convection, the characteristic morning low-level jet tends to be stronger and higher (at about 1 km height) than on days with shallow-only convection. The stronger low-level jet produces more VWS that induces more mechanical turbulence within the morning BL. Such added turbulence accelerates the formation of the convective BL and later invigorates cloud growth.

In combination with the mentioned processes, VWS can also help control aerosol-cloud interactions (Fan et al., 2009). The cloud tilting due to VWS, together with smaller droplets in polluted clouds, combines to inhibit cloud growth by enhanced evaporation rates. The core dilution will also affect aerosol-cloud interactions because aerosols tend to produce contrasting effects at the cores or margins of the clouds. Altaratz et al. (2014) provide a review of such contrasting effects, noting that the so-called cloud invigoration by aerosol pollution is related to cloud core processes. VWS can then counteract the invigoration mechanism by limiting the cloud core development.

In this study, we will quantify the effects of VWS on the physical dimensions of the clouds, their cores, and margins over the Amazonian region. We use the Dutch Atmospheric Large Eddy Simulation (DALES, Heus et al., 2010; Ouwersloot et al., 2017) model to simulate shallow cumulus clouds with high resolution, enabling the distinction of the cores and margins even in relatively small clouds. Novel in our research is the quantification of individual cloud, core, and margin dimensions as a function of VWS, which is enabled by a tracking algorithm. This quantification is also done as a function of the cloud lifecycle. The end goal is to help understand the direct VWS effects on the development of shallow clouds in the Amazonian

region, paving the way to further studies of the indirect effects. This will help constrain the environmental controls of Amazonian shallow cloud properties, which are still not well represented in global climate models (Lintner et al., 2017).

In Section 2 we present the model description, the experiment designs, and the methods used to analyse model output. Section 3 presents the main findings, followed by the summary and discussion in Section 4.

## 2 Methodology

### 2.1 Model description

This study uses the Dutch Atmospheric Large-Eddy Simulation (DALES) model in version 4.1 to perform idealised high-resolution simulations of shallow cumulus fields over the Amazonian region. This model has been used in many types of applications since its initial development as a tool to study the boundary layer and boundary layer clouds (Cuijpers and Duynkerke, 1993; Heus et al., 2010). Such applications include gas-phase chemistry (Vilà-Guerau de Arellano et al., 2011) including the influence of shallow cumulus (Vilà-Guerau de Arellano et al., 2005), aerosol-cloud interactions (de Bruine et al., 2019), and surface-cloud interactions in this case specifically for the Amazonian region (Vilà-Guerau de Arellano et al., 2020). DALES uses second-order central differences to solve most of the governing equations, except for the advective terms that use the methodology of Piacsek and Williams (1970). The radiative transfer is solved using the techniques described in Fu and Liou (1992) and Fu et al. (1997).

The code has been through significant recent improvements. The recent addition of explicit aerosol-cloud interactions is especially significant for this study (de Bruine et al., 2019). In short, the bulk microphysics scheme used in DALES (Seifert and Beheng, 2006) is updated, which previously used a fixed droplet number concentration combined with the classical supersaturation adjustment (Sommeria and Deardorff, 1977) for cloud liquid water. The new scheme prognoses cloud liquid water and droplet number concentration based on explicit treatment of the aerosol activation process following the κ-Köhler theory (Petters and Kreidenweis, 2007). The aerosol population is characterised by two lognormal size distributions, following the pristine rain forest (PR) conditions as defined in Pöhlker et al. (2018). This aerosol population is characterised by two modes at 69 nm and 157 nm, with concentrations of 162 $cm^{-3}$ and 86 $cm^{-3}$, respectively. The normalised standard deviations are 0.46 and 0.44, respectively. The hygroscopicity parameter κ for the two modes is 0.12 and 0.18, respectively.

The de Bruine et al. (2019) scheme has since been further updated to incorporate a prognostic relation for supersaturation:

$$\frac{\partial S}{\partial t} = dS_{macro} - \gamma S \tag{1}$$

Where $dS_{macro}$ represents the thermodynamic supersaturation source (linked to updraughts) and $\gamma S$ is the supersaturation sink related to cloud microphysics. Assuming a constant supersaturation sink coefficient γ within the timestep, Equation 1 can be solved to yield:

$$S(t + \delta t) = \frac{1}{\gamma}\exp(-\gamma\,\delta t)(dS_{macro}(\exp(\gamma\,\delta t) - 1) + \gamma S(t)) \tag{2}$$

The supersaturation sink coefficient ($\gamma$) is calculated as a proportion of the total droplet surface area obtained from the second moment of the cloud droplet size distribution. The Seifert and Beheng (2006) bulk scheme utilises a gamma size distribution for cloud and rain in the form:

$$N(x) = Ax^\nu \exp\left(-\lambda x^\mu\right) \tag{3}$$

Where $x$ is the droplet mass, and $A$ and $\lambda$ are proportional to the number concentration and mean droplet mass, respectively. For cloud droplets, $\nu$ and $\mu$ are both constant and equal to 1. Converting Equation 3 from droplet mass to diameter ($D$) and integrating, the total droplet surface area is given by:

$$H = \int_0^\infty \pi D^2 N(x(D))x'(D)dD = \left(9\pi \frac{q_c^2}{N_c^2 \rho_w^2}\right)^{\frac{1}{3}} \Gamma(8/3)N_c \cong 4.58\left(\frac{q_c}{\rho_w}\right)^{\frac{2}{3}} N_c^{\frac{1}{3}} \tag{4}$$

Where $q_c$ and $N_c$ are the cloud droplets specific liquid water mass and number concentration, respectively, $\rho_w$ is the water density, and $\Gamma$ is the gamma function. From this, $\gamma$ is then calculated as $\gamma = \alpha H$. By performing multiple tests, $\alpha = 50$ m$^{-2}$ kg is chosen because it produces realistic droplet number concentrations as compared to the aerosol number concentration.

**2.2 Experiments' description**

We perform six simulations, each changing only the vertical wind shear (VWS) of the initial profiles and large scale forcing. For all cases, the horizontal resolution is 50 m with a domain size of 21.6 km x 21.6 km and periodic boundary condition. The vertical domain goes up to 5 km, with a resolution of 20 m. This results in a 3D grid with dimensions of 432x432x250. The damping layer starts at 4 km height to avoid mass loss across the top of the domain. The time step is adaptive but with a maximum of 1 s, respecting the Courant-Friedrichs-Lewy (CFL) criterion. The output is done every 2 minutes, with averaged vertical profiles as well as 3D fields of selected variables mostly related to clouds, aerosols, thermodynamics, and turbulence. The simulations are centred at a latitude of -2.6091° and a longitude of -60.2093°, corresponding to the central Amazon. The total simulation time is 12 hours, from 6:00 to 18:00 in local time, with prescribed daily cycle of surface fluxes, as shown in Figure 1. Such profiles represent an idealised location consistent with the latitude and longitude of the domain centre on September 10[th] and for a tropical rainforest with Bowen ratios mostly between 0.1 and 0.4.

The input profiles of water vapour specific humidity $q_v$, potential temperature $\theta$ and the zonal and meridional wind components $u$ and $v$, respectively, are presented in Figure 2. The large-scale wind forcings are the same as the initial conditions for the respective runs and remain unchanged throughout the simulations. We have run DALES with six different characteristics of the input wind profiles but with $q_v$ and $\theta$ profiles remaining unchanged. The $q_v$ and $\theta$ profiles are based on radiosonde measurements performed during the GoAmazon2014/5 campaign (Martin et al., 2016) and they represent a typical dry season condition with predominant shallow cumulus formation. They were determined following Vilà-Guerau de Arellano et al. (2020).

The control simulation has no wind shear and is referred to as NS. For this simulation, the wind components are fixed to $u = -7$ m s$^{-1}$ and $v = -1$ m s$^{-1}$, matching the predominantly southeasterly winds of the region. To simulate different levels of wind

shear, we apply a linear increment in the initial and large-scale wind speed forcings throughout the whole vertical domain. Moderate VWS conditions (MS in Figure 2) are simulated by applying 1.2 m s$^{-1}$ km$^{-1}$ wind speed increments without changing the wind direction. High VWS conditions (HS in Figure 2) have twice as high wind speed increments (2.4 m s$^{-1}$ km$^{-1}$). According to the observational study of Chakraborty et al. (2018), the wind shear in MS and HS corresponds to values below the 33% and around the 50% percentiles for the Amazonian region, respectively. While our values may seem small, they represent a 6 m s$^{-1}$ and 12 m s$^{-1}$ total wind speed increment from 0 km to 5 km. Because we increase the wind speed linearly with height, higher VWS shear values could result in unrealistic wind speeds at 5 km. In addition to the NS, MS, and HS runs, we also perform additional tests by adding rotation to the VWS. This is done by applying a 90° clockwise rotation between 0 km and 5 km in the initial and large-scale forcings, also changing linearly with height. Those runs are referred to as NSR, MSR, and HSR and have the same wind speed vertical profiles as NS, MS, and HS, respectively, with the added directional shear. Throughout the simulations, the vertical profiles of wind speed and direction remain unchanged in the free troposphere, but they do suffer deceleration within the boundary layer. Overall, the wind speed within the boundary layer is up to 1 m/s higher in HS and HSR as compared to NS and NSR.

## 2.3 Analysis of model outputs

Given the relatively large number of clouds, we focus our analyses on the statistical properties of the cloud fields instead of individual case studies. We use two different approaches to analyse the outputs, which consist of domain averages of variables of interest and characterization of groups of clouds via a cloud tracking algorithm. For each approach, the data are also often analysed in terms of the cloud cores and margins. Heiblum et al. (2019) present three different methodologies to classify the convective cloud cores, which consist of finding pixels with either positive vertical velocity, supersaturation over 0%, or positive buoyancy. Here we will use the most restrictive definition of the cloud core, consisting of a combination of all three conditions concomitantly. Because we quantify the magnitudes of the clouds, their cores, and their margins, this definition has a significant impact on our results. Changes in the core definition will then directly affect those magnitudes. Our restrictive definition of cloud cores implies that we are focusing on the most intense part of the cloud to further differentiate it from the margin statistics. Similarly, the margins are defined as the pixels with concomitantly negative vertical velocity, subsaturated conditions, and negatively buoyant air.

### 2.3.1 Cloud Tracking

We use the Forecasting and Tracking the Evolution of Cloud Clusters (ForTraCC) (Vila et al., 2008) algorithm to track clouds. Initially developed to track cloud clusters in satellite imagery, the code has been further developed to work with radar data (Queiroz, 2009). Specifically, the code was adapted to read and track rain cells in Constant Altitude Plan Position Indicator (CAPPI) files. Such files contain a two-dimensional grid with radar reflectivity values, where the algorithm tracks shapes with reflectivity values higher than a predetermined threshold. An example of use can be found in (Cecchini et al., 2020). This adaptation made it easier to use the tracking algorithm in other applications such as modelling. Here we produce

two-dimensional fields of the cloud Liquid Water Path (LWP) between 800 m and 1500 m height to use as input to ForTraCC. The algorithm then tracks the clouds based on a LWP threshold of 0.6 g m$^{-2}$. With a vertical resolution of 20 m, this threshold corresponds to 3 vertical pixels with 0.01 g m$^{-3}$ of cloud Liquid Water Content (LWC), a common threshold to identify clouds in numerical models. Beyond the LWP threshold, ForTraCC is setup to track clouds that have at least 10 pixels with LWP over the threshold to minimise noise. This means that the minimum trackable area is 25000 m$^2$, which

corresponds to a circle with an approximate diameter of 180 m. Therefore, some of the initial and final cloud life cycle beyond this length scale is missed in the tracking, especially considering the 2-minute interval of the model output and the 50-m horizontal resolution. It is important to take those limitations into consideration when analysing the results presented here.

  The tracking works based on the overlapping area of the trackable pixels between two different time steps. In each time step,

the shapes that maximise such an overlap area with the previous time step will be considered to be part of the same clouds. Therefore, the speed of the moving shapes is important in this process because fast-moving clouds can reduce or even eliminate such overlap. For instance, the wind components of $u$ = -7 m s$^{-1}$ and $v$ = -1 m s$^{-1}$ defined in the NS run represent a total wind speed of approximately 7.1 m s$^{-1}$. At such a speed, the shapes could move 850 m in two minutes, which is greater than the minimum trackable area. However, the DALES simulations have a moving domain to minimise the speed of cloud

displacement. We have set the domain speed to match the wind speed at 1 km height, which is close to the cloud base. This drastically reduces cloud displacement and maximizes the overlapping areas, allowing ForTraCC to track the shapes.

  Figure 3 presents an example of the tracking of clouds in the entire domain for the NS run. The colours represent the LWP (between 800 m and 1500 m) field at 12:00 local time, and the rectangles show up to four different time steps of the tracked clouds. The rectangles cover the area defined by the largest cloud dimensions in the West-East and South-North directions.

Black rectangles represent the current time step and progressively lighter grey rectangles represent earlier time steps in order (i.e., the lightest rectangles are three time steps earlier than the current time shown in Figure 3). This confirms the area overlap between the time steps of the clouds, allowing the establishment of the Lagrangian database.

  The area enclosed by the rectangles represents the horizontal coordinates of the individual clouds and will increase and decrease as the clouds develop. From those horizontal coordinates, we define the 3D field of individual clouds by using the

215 altitude interval from the surface to the cloud top, removing multi-layered clouds. Multi-layered clouds are defined when there are non-cloudy pixels between two cloudy pixels in the same column according to the 0.01 g m$^{-3}$ threshold. Whenever multi-layered clouds are detected, we only use the bottom one for the remainder of the calculations. Cloud top height (CTH) is then obtained as the maximum altitude within the rectangles where LWC > 0.01 g m$^{-3}$. As such, for every trackable cloud, we have a moving 3D volume of pixels for which we store properties like LWC, rate of change of LWC ($\Delta$LWC), cloud

droplet number concentration $N_c$, updraught speed $w$, buoyancy $B$, among others. This forms the basis of the cloud tracking statistical results shown here. Additionally, we classify each pixel in the 3D volumes as cloud core or margin (or none) based on the criteria discussed previously. This allows quantification of cloud core and margin properties for the individual clouds as well.

Note that in Figure 3 there are no tracked rectangles close to the borders of the domain. This is because we explicitly filter out clouds that at any point in their life cycle had one or more pixels at the boundaries. The main reason for such a filter is the periodic boundary condition, where a cloud moving through one extremity would appear in the opposite one. Since we define a new cloud as the first time it attends to the ForTraCC tracking criteria, the periodic boundaries could interfere with the statistics. For instance, if an already mature cloud moves through the boundary, it would later be defined as a new cloud in the opposite extremity. Therefore, its initial characteristics would be different than clouds that form and dissipate within the borders.

It is possible that two different clouds have overlapping 3D fields if they are within close proximity to each other. To minimise such effect and avoid counting the same cloud multiple times in the overall statistics, we apply a filter to remove other clouds from the 3D volumes. This is done from the pixel classification done within ForTraCC. The algorithm provides an identification number for each cloud, which is then used to eliminate other clouds from the 3D volumes. Whenever a different cloud is identified within the 3D volumes, we set all of its variables (e.g., LWC, $N_c$, etc) to 0 only for the 3D volume, including all vertical pixels above the smaller shapes. In this way, only the tracked cloud is left in the volume. Note that the Lagrangian cloud database is stored separately from the domain variables. Therefore, this process does not affect the domain-wide characteristics.

The ForTraCC algorithm also detects the occurrence of mergers and splits in the tracked shapes. Mergers occur when two or more tracked shapes have their areas joined at some point in their life cycle. Splits are defined when one shape in one time step is split into two or more shapes in a subsequent time step. In such a case, ForTraCC continues tracking the split area with the largest overlap with the original shape. The shapes with smaller overlaps can start a new cloud life cycle if they adhere to the LWP threshold and minimum area requirements. The occurrence of mergers and splits is stored as flags together with the 3D volumes, so we can identify them in the statistics. Here we will mostly discuss the cloud characteristics independently if they went through mergers or splits throughout their life cycle. The main reason is to increase the statistical robustness since completely isolated clouds correspond to less than half of the total cloud counts. Additionally, the increased wind shear tends to increase merger and split occasions because or larger horizontal areas overall. Therefore, for the higher-VWS runs, this effect would be further intensified.

# 3 Quantifying VWS effects on the physical characteristics of clouds

## 3.1 Overall characteristics of the experiments

Here we present a brief description of overall characteristics of the experiments and how the reference run compares with observations. Firstly, Figure 4 presents the evolution of domain-averaged LWP (Figure 4a), the 99% percentile of domain-wide CTH (Figure 4b) and the domain-averaged rainwater content at the surface ($RWC_{sfc}$) (Figure 4c). The time series have been smoothed by 30-minute moving averages for clarity. It shows that the cloud field properties are consistent with shallow convective clouds, with averaged LWP mostly below 200 g m$^{-2}$, aside from the peak at 14:30 local time in the HS

simulation. This peak in LWP is due to one deeper cloud that has no equivalent in the runs with lower VWS. This cloud also brings the 99% percentile of CTH to above 4000 m and is the only one that generates significant precipitation at the surface. Note that heights above 4000 m are already inside the damping layer, indicating that the cloud could have been even deeper if the domain top was higher. Aside from that, Figure 4 shows that the cloud field is mostly representative of non-

precipitating shallow cumulus clouds.

Figure 5 presents a comparison between observed and simulated profiles of LWC alongside the cloud cover (CC) distribution with height. The observed LWC is taken from flight AC09 of the ACRIDICON-CHUVA experiment (Wendisch et al., 2016; Machado et al., 2018), which consisted of a cloud profiling mission over the Northern Amazon on September 9[th], 2014. We have limited the analysis to the first 3000 m to maximise the number of datapoints both from the simulations

and from the aircraft data. The altitude of the observed data has been reduced by approximately 500 m, so the minimum observed and simulated LWC are at almost the same altitude. The observed LWC is shown as round markers representing the average of different flight legs (constant altitude) as well as the standard deviation (error bars). The observations were collected between 11:30 and 14:50 local time. The simulated LWC is shown in the green continuous line and the shaded area around it. For every model time step between the 11:30-14:50 time interval, we obtain the median and interquartile range of

LWC as a function of altitude. This is then averaged over the period to get the pattern shown in Figure 5. Alongside this comparison, we also show the averaged profiles of cloud cover for the NS, MS, and HS runs (top horizontal axis).

Figure 5 shows that the vertical profile of LWC is within the range of the observations at almost all levels between 500 m and 3000 m. There are some exceptions at altitudes close to 2300 m and 3000 m where the observed LWC is larger than the simulations. This is probably explained by the fact that the observed clouds were actively producing precipitation, while the

simulated ones are non-precipitating. In turn, this could also explain the close LWC values between observations and simulation below 2000 m. In terms of cloud cover, Figure 5 shows the expected result of increasing CC as VWS increases. For a fixed altitude, the average difference of CC is up to about 2% between NS and HS.

## 3.2 Representativeness of the tracking algorithm

Before going into detail about the results from the cloud tracking algorithm, we present a contextualization of its representativeness of the overall cloud field. Figure 6 presents the normalised histogram of LWP for all pixels in the simulation domain (continuous black lines) for all six runs and all output time steps between 11:00 and 18:00 local time. Such curves represent the overall characteristics of the cloud field in the simulation domain. We replicate this calculation only for the pixels tracked by ForTraCC to infer any possible bias towards specific LWP ranges. The dashed lines represent

all pixels tracked, while the dot-dashed lines represent only the pixels within clouds with no merger or split detected throughout their life cycle. For simplicity, the latter will be referred to as "isolated" clouds.

When the different lines in Figure 6 have close proximity to each other, it means that such an LWP range has approximately the same weight in the statistics associated with the whole domain or with the tracked clouds. This is the case for the LWP <

50 g m$^{-2}$ range, meaning that the smaller clouds have approximately the same representativeness in the whole-domain statistics as compared to the tracking algorithm. However, there is an overall trend of reduced representativeness of large values of LWP. Large LWP are likely related to deeper clouds, which are also likely larger horizontally and have longer lifecycles. For LWP > 50 g m$^{-2}$, the normalised counts of all tracked pixels have a diminishing representativeness of the whole-domain normalised counts, with this effect being even stronger for isolated clouds. Therefore, the statistics of the tracking algorithm have an overall bias towards smaller clouds as compared to whole-domain statistics, but this difference is significantly reduced if all tracked clouds are used. The main reason for this bias is the exclusion of clouds crossing the boundaries. Since the larger and deeper clouds have a higher chance of crossing the boundaries during their lifecycle, some of them can be excluded from the tracking results.

Some previous studies applying tracked algorithms to model outputs focus mostly on isolated clouds (e.g., Heiblum et al., 2019), which is preferred when analysing the properties of individual clouds. Here, we are mostly interested in the overall cloud field metrics, and all tracked clouds will be analysed together regardless of merger and split occurrences. This decision is motivated by the results shown in Figure 6. If only isolated clouds were used, the VWS effects would only be valid for relatively shallow and short-lived clouds. By keeping all clouds, most of the cumulus field can be analysed together and the statistics get more statistically robust by increasing sample size.

This comparison shows that we can use ForTraCC in combination with domain averages to analyse different aspects of the cloud field. The larger clouds usually dominate the cloud-related statistics in the domain averages, while the Lagrangian dataset from ForTraCC allows a more detailed study of the baseline shallow cloud field. Here this distinction between the baseline shallow cloud field and the deepest clouds in the domain is important for the remainder of the analyses. The tracking results from ForTraCC are skewed towards the baseline shallow cloud field since such clouds tend to be smaller and short-lived. On the other hand, the deepest clouds develop within and are supported by the underlying shallow cloud field. We argue that the properties of the underlying shallow cloud field have a significant effect on the development of the deepest clouds. Therefore, both the underlying shallow cloud field and the deepest clouds will be discussed separately from now on.

For example, Figure 7 shows the daily cycle of total domain water for all six simulations, calculated as the 3D integration of the cloud and rain liquid water content over every pixel in the entire domain. This figure shows that the total domain water changes significantly with VWS, even though the relation is not straightforward. The time series between NS and MS are relatively similar, with only as 3% difference between their total water amount peaks close to 14:00 local time. The patterns throughout the day are also similar, with mostly 10-min gaps between the local maxima and minima. However, HS drastically differs from those patterns, having a peak total water about 3 times higher and close to 15:00. This is because this run generated the largest cloud in this study, as shown in Figure 4. As will be discussed later, we argue that this largest cloud was supported by a more favourable shallow cloud field under high VWS. Interestingly, the directional shear interacts in different ways with the total domain water and with their non-directional wind shear counterparts. For NS and HS, the addition of wind rotation reduces the total water peak by approximately 20% and 40%, respectively. However, for moderate

VWS conditions, the directional shear increased the peak by about 30%. Therefore, the VWS seems to interact in nonlinear ways with the cloud field even under idealized simulation conditions.

### 3.3 VWS effects on the cloud's external and internal dimensions

The overall diameter of the individual clouds is estimated by calculating the equivalent diameter of a circle with the same area as the tracked shapes. With this calculation, the effects of VWS on the horizontal cloud dimension are quantified. Figure 8 shows the resulting probability density functions (PDF) of cloud equivalent diameter for the six runs for 100-m intervals between 100 m and 1800 m. Those size intervals represent most of the clouds in the domain, and each datapoint in Figure 8 represents at least 130 clouds. The PDFs are shown in panel a) as functions of the central diameter of the intervals, while the

ratios of the PDFs are shown in panel b) (having the NS simulation as the reference) for comparison. The numbers in parenthesis in the legend of Figure 8a are the total cloud count for the respective runs.

Firstly, there is a notable decrease in cloud count as VWS shear increases. Between the NS/NSR and the HS/HSR runs, there is a reduction of about 900-1000 clouds overall. Such a reduction is related to enhanced cloud clustering similar to the results from Yamaguchi et al. (2019). Additionally, the cloud tilting due to VWS favours the evaporation at the cloud margin,

potentially causing the smallest clouds to fall off the ForTraCC minimum area criteria. Figure 8 shows that there is a decrease in the frequency of occurrence of clouds smaller than 400 m under high VWS conditions. Such a reduction reaches up to about 20% in frequency, as evidenced in Figure 8b. This could be explained by both the enhanced evaporation of smaller clouds, as well as by the overall increase in cloud area caused by the enhanced mechanical wind forcing. Clouds larger than 800 m are progressively more frequent with increased VWS, up to a maximum increase of 190% in frequency.

Such an increase also leads to larger values of cloud cover (Figure 5).

While the sheared environments produce larger clouds horizontally, the tracking results evidence that they are also shallower. Figure 9 presents the same analysis as Figure 8, but for cloud depth. Here we define cloud depth as the height difference between cloud base (lowest cloud pixel height with LWC $> 0.01$ g m$^{-3}$) and cloud top (highest cloud pixel height with LWC $> 0.01$ g m$^{-3}$). In Figure 9, there is a notable decrease in the frequency of occurrence of clouds deeper than 1200

345 m for the sheared runs, with the opposite for shallower clouds. This indicates that the infrequent deeper clouds generating the highest total domain water peak shown in Figure 7 occur concomitantly with a weakening of the smaller and more numerous clouds in the field. Therefore, it shows that whole-domain properties such as the total domain water are heavily influenced by infrequent and large clouds, as should be expected.

More details about the VWS effect on the clouds can be analysed by constraining the cloud core and margin characteristics.

With the tracking algorithm, we calculate the core and margin dimensions as a function of both cloud lifetime and height. To combine multiple clouds with different depths and lifetime duration, we perform normalizations in the height and lifetime, taking into account the cloud base and top heights and the initial and last time steps of the clouds. The relative cloud depth is set to 0 at cloud base and 1 at cloud top, scaling linearly with height between those extremes. Similarly, the relative lifetime ($t_{rel}$) is set to 0 and 1 at the initial cloud detection and at the last tracked time step, respectively. Figure 10 shows the

averaged dimensions of the clouds, cores, and margins as a function of the relative depth and lifetime of the clouds. As before, we report the dimensions as the equivalent diameter of an equivalent-area circle. The relative life cycle is split into three periods, roughly corresponding to the formation, maturation, and dissipation stages. The relative depth is split into 5 intervals centred at 0.1, 0.3, 0.5, 0.7, and 0.9. The variability around the averages is not shown for figure clarity, but the standard deviation represents about 80% of the averages close to cloud base, reducing to 50% close to cloud top.

Figure 10a shows that the clouds have averaged dimensions between 125 m and 550 m at different vertical levels, with maximums close to 0.3 relative depth. This relative depth corresponds to the maximum dimension of the cores (Figure 10b) and margins (Figure 10c). The cores tend to be larger at the early and mid-lifecycle stages, reducing in size at the dissipating stage. On the other hand, the cloud margins tend to be similar in dimensions at the dissipating stage as compared to the early lifecycle. Overall, the cloud cores are about 60-150 m larger than the cloud margins at a given relative depth according to our methodology. This difference tends to be larger close to cloud base but lower close to cloud top. We note that the sum of the core and margin lengths does not necessarily add up to the cloud length because of the different samplings involved. In fact, most of the points shown in Figure 10b,c present larger sums of core and margin lengths than the cloud dimensions in Figure 10a. The reason is the restrictive classification of the core and margin that requires updraught speed, buoyancy, and supersaturation to be all positive (core) or all negative (margin). Therefore, this allows occurrences of cloudy pixels with no core or margin classification. Overall, cloudy cross sections that have no core or margin pixels tend to be smaller than those with core or margin pixels. This results in the cloud lengths in Figure 10a having a bias towards lower values.

The VWS effect on the clouds is mostly consistent throughout the clouds' relative depth and lifecycle. On average, VWS increases the dimensions of the cloud cores and margins by up to 40 m each. This totals an increase in horizontal cloud dimensions by up to 80 m. The increase in average cloud dimensions is almost linear with the vertical wind speed shear, while the added rotations present a weaker effect. Here we note that the effects on the internal cloud dimensions are close to the resolution utilized in the simulations. Therefore, it is desirable to do simulations with finer resolutions in the future. However, the proportionality of equivalent diameters with VWS between NS, MS, and HS indicates consistency in this relationship.

While Figure 10 shows increased cloud dimensions due to VWS, it does not reveal the relative magnitudes of the core and how it changes with VWS. Figures 11 and 12 provide calculations of relative magnitudes in terms of horizontal area (Figure 11) and volume (Figure 12). Figure 11 is similar to Figure 10, but the profiles represent the average fraction of the cloud core area (i.e., the ratio of the core and cloud areas) in horizontal slices through the clouds. Figure 11a shows the actual fraction values, while Figure 11b shows the differences between the MS/HS runs and NS. Since the directional shear did not have as much impact in the cloud dimensions as compared to the wind speed shear, the runs NSR, MSR, and HSR are omitted in Figure 11 for clarity. The standard deviation of the area fraction represents approximately 70% of the cloud-base averages, decreasing to 50% towards cloud top.

Figure 11a shows that the cores represent between 25% and 45% of the horizontal area of the clouds, and can change with lifetime, height, and VWS. As expected, the relative area of the cores tends to decrease towards the dissipating stage of the clouds. In our methodology, this reduction is approximately 3% on average. Note that even in the dissipating stages (yellow curves in Figure 11), the core areas still represent more than 25% of the cloud area. As mentioned before, this results from the minimum area requirement for the tracking algorithm as well as the broad relative lifetime intervals used in Figure 11. In all lifecycle stages, the core area fraction tends to increase with relative depth, consistent with a buoyant core location close to cloud top.

Figure 11b shows that moderate VWS conditions have a small effect on the core area fraction, with averaged changes mostly within 1% between MS and NS. On the other hand, high VWS presents an overall reduction of 3% in the core area fraction. This reduction is of the same magnitude as the reduction throughout the tracked lifetime of the clouds, as shown in Figure 11a (differences between the blue and yellow curves for all runs). Therefore, it is possible to conclude that high VWS tends to both increase the absolute dimensions of the clouds and their cores as well as significantly reduce the core area fraction. This is consistent with the larger cloud sizes shown in Figure 8, with the smaller cores being responsible for the shallower clouds under high VWS as shown in Figure 9.

Figure 11 shows the overall volume and mass fractions of the cores as functions of the clouds' lifecycle and VWS. Every curve in this figure represents the averaged core volume (Figure 12a) and mass fractions (Figure 12b) of clouds with the same duration (shown in colours). Here the duration is simply calculated from the number of output time steps where a specific cloud was detected by ForTraCC. If the cloud appears in two consecutive output time steps, it is considered to have a two-minute duration. For three time steps, the duration is 4 minutes and so on. The horizontal axes represent the relative time of the clouds, in minutes, i.e., representing each time step in their lifecycle. There are at least 38 clouds in every duration category, with larger numbers for shorter durations. The volume and mass fractions standard deviations are relatively similar for the time steps of clouds sharing the same duration. For the volume fraction, the standard deviations reach up to approximately 0.15. For mass fractions, they have values up to 0.22 approximately.

The evolution of the core volume and mass fractions is consistent with the expected convective lifecycle, with growing fractions in the earlier stages followed by shrinking cores during dissipation. The core mass fractions are approximately 0.10-0.15 higher than the volume fractions because of the supersaturated conditions within the cores that cause droplet growth and, therefore, mass gain disproportionally to the volume. The consistent relation between the volume and mass fractions and cloud duration is notable and expected. The more core-dominant the clouds are, the longer they take to dissipate.

For clouds lasting 6 minutes or more, Figure 12 shows a consistent tendency of reduced core volume and mass fractions for higher VWS conditions (with only a few exceptions), predominantly close to and after the peak values. It is possible that the short-lived clouds do not present such a clear pattern because the variability is of the same magnitude as the averages. For longer-living clouds, the averaged fraction values tend to be greater than the standard deviation, and this coincides with the

definition of the pattern. Regardless, such a reduction of core volume and mass fractions is of the order of 0.05 and is consistent with the reduction of the area fractions as shown in Figure 11.

Beyond the magnitudes of the cores, it is also important to quantify the cloud characteristics within them. Most importantly, the amount of liquid water mass as well as the vertical motions. Figure 13 presents the bivariate histograms of $w$ and LWC for the cloud cores in the NS, MS, and HS runs irrespective of height. The curves represent the number of datapoints for $w$

and LWC intervals of 0.2 m s$^{-1}$ and 0.1 g m$^{-3}$, respectively. The coloured shapes and green continuous lines represent the NS run. The continuous magenta and dashed cyan lines represent the MS and HS runs, respectively. This figure shows that there is a clear relationship between the core convective strength and the amount of wind shear. There is a progressive trend of lower $w$ and LWC values as the wind shear increases. This can be seen by comparing lines representing the same number of datapoints in Figure 13. For instance, compare the lines representing 1000 datapoints for the different runs. The MS line

represents a domain closer to the low-$w$/low-LWC region of the graph as compared to the NS run. Similarly, the 1000-datapoint line for HS is even closer to such a region than MS. This is repeated for basically all isolines between 50 and 3000 datapoints.

This shows that the added wind shear constrains the $w$ and LWC datapoints within a relatively limited space, with overall reductions of $w$ and LWC. Therefore, VWS not only reduces the relative dimensions of the cores but also decreases the

convective intensity within them. Of course, such factors are correlated because weaker cores will result in smaller physical dimensions, but there is a noticeable and consistent effect of VWS on the cloud core properties. For a given isoline in Figure 13, its maximum LWC and $w$ are reduced by approximately 0.1 g m$^{-3}$ and 0.6 m s$^{-1}$, respectively, between NS and HS. Such a reduction will affect the water mass fluxes caused by the convective clouds, therefore changing how they affect the atmosphere in return.

For instance, the shallower but larger clouds with weaker cores in the HS run will result in different profiles of latent heat release and consumption, which will change how they cool or warm the air around them. This, in turn, will result in different impacts on the vertical temperature and humidity profiles, affecting the preconditioning of the atmosphere. With the aim of estimating the enhanced preconditioning, we have calculated the averaged profiles of the condensation/evaporation rates for all cloudy pixels in all runs. This is done by averaging the cloud droplet water tendency (ΔLWC) for all cloudy pixels in

every timestep. The result is a single vertical profile of ΔLWC for every timestep in all runs. To eliminate the contribution of rain formation to ΔLWC, we only perform the calculations for pixels with no rainwater (using the threshold of 0.001 g m$^{-3}$). Figure 7 evidences that the deepest cloud in HS occurred between 14:00 and 15:00 local time. Therefore the preconditioning mechanisms should be analysed before such a timeframe.

To estimate the ΔLWC effect on the vicinity of clouds, we integrate the averaged profiles of ΔLWC over time between

11:00 and 14:00 local time. This results in an averaged profile of water vapour release or removal from the atmosphere based on the evaporation and condensation within the clouds. The results are shown in Figure 14a, with the different runs represented by different line colours. Because of the pulsating nature of convective cloud development and the resulting changes to ΔLWC, the variability is very high. The standard deviation of ΔLWC prior to the time integration is about 1-2

orders of magnitude higher than the averages, due in large part to the variable cloud depths, lifetimes, and convective strengths in every time step. Nevertheless, Figure 14a is consistent with our previous results because there is an enhancement of the water vapour release with higher VWS due to the larger cloud areas as well as the relatively lower core fractions. There is more cloud water being converted into water vapour between 1500 m and 2500 m, as seen in Figure 14b, which shows the differences of the time-integrated $\Delta$LWC between the different runs (having NS as the reference). Figure 14b shows that higher VWS can lead to up to 0.3 g m$^{-3}$ more water vapour being released to the atmosphere on average due to the enhanced bulk evaporation of the clouds between 11:00 and 14:00.

The evaporation of cloud droplets decreases the air temperature ($T$) in the vicinity due to the latent heat exchange. Additionally, it also increases the relative humidity (RH) by adding more water vapour to the atmosphere. Conversely, water vapour condensation increases $T$ and reduces RH. We estimate such $T$ and RH tendencies based on the curves of Figure 14b, by assuming a latent heat of condensation $L_v$ of $2.5\times10^6$ J kg$^{-1}$ and a specific heat of air at constant pressure $c_p$ of 1005 J kg$^{-1}$ K$^{-1}$. The RH results are calculated after the temperature corrections, therefore taking into account both the temperature and humidity variations. The results are shown in Figure 14c,d. It shows that the enhanced evaporation with high VWS results in a temperature decrease of up to 0.8 °C at a height of 2000 m. Similarly, there can be a positive temperature tendency of up to 0.6 °C below 1500 m or above 2500 m, potentially due to the reduced number of clouds and their depths overall. The RH tendencies have opposite sign as the temperature tendencies and are mostly contained between -6% and +7%.

To investigate whether the estimated $T$ and RH changes would leave the local atmosphere more or less unstable, we calculate the corresponding convective available potential energy difference ($\Delta$CAPE) based on the curves of Figure 14c,d. Firstly, the temperature and humidity differences are converted into virtual temperature ($T_v$) differences. Then, this is used to calculate $\Delta$CAPE as:

$$\Delta CAPE = -\int_{z=0}^{z=5000} g\,\frac{\Delta T_v}{\overline{T_v}}\,dz \qquad (1)$$

Where $g$ is the acceleration due to gravity (fixed to 9.8 m s$^{-2}$ here), $\Delta T_v$ is the mentioned virtual temperature difference, and $\overline{T_v}$ is the baseline virtual temperature profile. This baseline profile is taken from the domain averages (cloudy and non-cloudy pixels) of the NS run. The negative sign is added because negative $\Delta T_v$ leads to an increase in CAPE for later cloud parcels developing nearby. The $\Delta$CAPE calculation provides the overall net effect of the $T$ and RH differences due to higher VWS conditions. The values of $\Delta$CAPE are 5.2 J kg$^{-1}$, -1.6 J kg$^{-1}$, 14.1 J kg$^{-1}$, 13.1 J kg$^{-1}$, and 18.1 J kg$^{-1}$ for the NSR, MS, MSR, HS, and HSR runs, respectively. Except for MS, all simulations result in increased CAPE due to the enhanced evaporation effect, suggesting that later clouds can be invigorated. Additionally, the added directional shear seems to affect the $\Delta$CAPE calculations more than the wind speed shear, which is surprising considering that the rest of the results shown here point to the stronger effect of wind speed shear in the cloud dimensions. This could be related to the increased cloud cover of the runs with directional shear. Those runs have 10% higher average cloud cover as compared to their respective non-directional shear runs (not shown).

While most ΔCAPE values are positive, their absolute values are relatively small, and it is inconclusive if they would be enough to generate the deepest cloud in the HS run. Of course, the ΔCAPE values represent an average throughout the cloudy pixels in the domain and the ΔLWC profiles have high variability, so there can be variations depending on the cloud cover in each subsection of the domain. The higher VWS not only increases the cloud cover overall, but it also tends to increase cloud clustering. Such groupings of clouds could potentially generate more instabilities in specific sections of the domain, invigorating the later clouds that form there even though the domain as a whole is not much more unstable. We note that our estimates of ΔCAPE represent the atmosphere in the immediate vicinity of clouds and therefore do not represent the whole domain. On average, the horizontal cloud cover reaches a maximum of about 2.4% at 1100 m (Figure 5), indicating that the effects of evaporation should be small across the entire domain. There are also other processes that can cause invigoration of clouds by VWS, including the intensification of turbulence at the boundary layer (Henkes et al., 2021). The identification of processes that can lead to cloud invigoration due to increased VWS in our simulations will be the subject of future studies.

## 4 Summary and discussion

This study analysed the effects of vertical wind shear (VWS) on the properties of cumulus cloud fields over Central Amazonia using idealized model simulations. The cloud fields were simulated by the Dutch Atmospheric Large Eddy Simulation (DALES) model with a domain size of 21.6 km x 21.6 km (horizontal, 50 m resolution) x 5.0 km (vertical, 20 m resolution). To this end, a suite of systematic numerical experiments was performed, differing only by the initial and large scale VWS forcing. The reference run had no wind speed or directional VWS and is referred to as NS. The vertical profiles of temperature and moisture represent a typical day during the dry season and are based on radiosondes (Vilà-Guerau de Arellano et al., 2020). Moderate (MS) and high (HS) vertical wind speed shear conditions were simulated by increasing the wind speed by factors of 1.2 m s km$^{-1}$ and 2.4 m s$^{-1}$ km$^{-1}$, respectively. Three additional runs were performed by adding 90° of wind rotation linearly between the heights of 0 km and 5 km to study the effect of directional wind shear. This rotation is added on top of the three different wind speed shear conditions. Such runs are referred to as NSR, MSR, and HSR.

A tracking algorithm allowed a Lagrangian examination of the cloud properties, enabling us to analyse how cloud properties are evolving in time and space. We find that increasing VWS leads to larger clouds horizontally while at the same time limiting their vertical development. The idealized simulations of a representative day with shallow, non-precipitating cumulus development in Amazonia show that clouds with an equivalent diameter (diameter of a circle with the same horizontal area) greater than 1000 m are between 1.2 and 2.0 times more likely under high VWS than with no shear. Conversely, cloud depths lower than 1000 m are about 1.2 times more likely under high VWS. The tracking algorithm revealed that the increase of the clouds' horizontal dimensions is consistent throughout their vertical structure as well as their lifecycle. On average, the equivalent diameter of the clouds is increased only by up to 80 m by VWS, where the cloud cores and margins are increased by up to approximately 40 m each. This is of similar order to the horizontal resolution used in this

study, indicating that small grid cells are desirable to study shallow clouds in more detail. Nonetheless, the increase in cloud dimension was shown to be proportional to VWS, where the MS run produced slightly larger clouds than NS and HS produced slightly larger clouds than MS. This indicates that there is consistency in the VWS effect in the cloud dimensions even though the averaged values are small and comparable to the grid size used.

While the cloud cores tend to be larger under higher VWS, their relative magnitudes change. Under high VWS, the core horizontal area fraction (i.e., the ratio of the core area and the cloud area) is reduced by about 0.025, which represents approximately 6.5% of the overall core area fraction average in the NS run. In contrast, the cores of clouds developing in sheared environments presented lower updraught speed $w$ and cloud droplet liquid water content LWC overall, which resulted in a reduction of up to 0.05 in the cores' mass and volume fractions. In relative terms, having NS as reference, this represents a relative reduction in the range of 10%-25% approximately. It is important to stress that the sensitivity of cloud properties to the vertical wind speed shear is greater than the directional shear.

Our findings are consistent with previous studies suggesting that VWS weakens convective clouds and their cores by tilting them vertically (e.g., Helfer et al., 2020; Drueke et al., 2021). However, despite having shallower cumulus clouds with overall weaker cores, we find that the few deepest individual clouds simulated occurred under sheared environments. More specifically, the HS run have simulated the deepest cloud that resulted in a threefold increase in the total domain liquid water mass as compared to NS. This apparent contrast, where most clouds in the field are inhibited but a few may invigorate, can indicate a non-linear aspect of the VWS-clouds interactions. Changes of the early cumulus cloud field properties can feedback into the thermodynamics of the atmosphere with consequences for the later development of new clouds, a process known as preconditioning (Neggers et al., 2007). We have investigated one aspect of this possible feedback by analyzing the total cloud field evaporation and its theoretical effect on the temperature and humidity vertical profiles. We have shown that the mostly larger but shallower clouds of the high-VWS runs result in larger total evaporation in the layer ranging from 1500 to 2500 m height. This, in turn, destabilizes the atmosphere by decreasing the temperature and increasing the humidity of such layer. The result is a local increase in the convective available potential energy (CAPE) that can affect the formation of clouds developing later in the region. While the domain-averaged CAPE increase is relatively low (up to about 18 J kg$^{-1}$), it could vary significantly depending on the cloud cover of specific subsections within the domain. The subsections with more cloud development during the morning will increase the chances of developing a deeper cloud downwind in the afternoon. This localized preconditioning may be one aspect of the increased cloud clustering due to VWS as discussed in previous studies (e.g., Yamaguchi et al., 2019). The VWS influence could also represent an increase in the preconditioning effect where shallow clouds supply the low-to-mid tropospheric humidity needed for deeper convection.

Two potential indirect effects of VWS on the cloud field development are related to the convective boundary layer formation. Recently, Henkes et al. (2021) analysed 2 years of observations of the boundary layer (BL) conditions previous to shallow-only (ShCu) convection and shallow-to-deep convective transition (ShDeep) in the Amazonian dry season. In their conceptual model, the wind shear close to the top of the BL plays an important role in determining the morning BL evolution and setting up the conditions for deeper convection. The authors note that the low-level jet commonly formed right over the

morning BL is usually stronger and at a higher altitude on ShDeep days as compared to ShCu. A stronger VWS increases the entrainment rates on the top of the BL (Pino et al., 2003), accelerating its growth. Indeed, our calculations show that the BL height is about 150 m higher in the HS run as compared to NS. Helfer and Nuijens (2021) suggest that increases in the VWS within the BL separate the precipitating downdrafts and the updraughts below cloud base that support cloud formation. This process, together with a strengthened convective BL, could help explain the formation of deeper clouds in sheared environments.

We therefore suggest that VWS influences the cloud field in both direct (by tilting and stretching the clouds) and indirect ways (by feedback mechanisms from cloud evaporation and BL formation). In our study, we have focused mostly on the quantification of the VWS in the cloud properties (i.e., direct effect), which shows an overall inhibition of convective development under sheared environments. However, both our simulations and the results from Henkes et al. (2021) point to a more complex relationship between VWS and convective cloud formation and growth in the Amazon. Of course, the natural occurrences of VWS over the region are not linear as in our simulations. Often, the VWS in the first few kilometres of the atmosphere manifests itself as a low-level jet. This type of phenomenon has a particular vertical structure where there is a specific height with peak wind speeds (usually around 0.5-1.0 km), which could have different effects on the cloud field. Other types of VWS, such as the low-level and deep VWS mentioned in Chakraborty et al. (2018), can also occur at the same time, adding more complexities to the analysis.

The idealized nature of the simulations performed here do not allow a concrete conclusion on whether VWS can support the formation of deeper clouds due to enhanced preconditioning and intensification of the morning boundary layer or if this is a sampling problem. This result can be statistical noise and appears due to non-infinite sampling comparisons. In addition, Vilà-Guerau de Arellano et al. (2020) have shown that the forest and cloud formation are intrinsically linked over the region. A natural continuation of this study is to replace the use of prescribed surface fluxes with a more realistic surface-atmosphere interaction scheme to represent the coupling between the Amazonian clouds and the dynamics of the soil and rainforest surface turbulent fluxes. Nevertheless, the advantage of using idealized simulations is that it enables us to break down the complexity and systematically analyse individual processes. The convective cloud dimensions were changed marginally by modifying the VWS initial and large-scale forcing conditions. Nevertheless, this has led to changes in the cloud field as a whole, even allowing the formation of a deeper and precipitating cloud under high shear conditions. This indicates that VWS is an important aspect of cloud formation over the region. In particular, the direct VWS effect inhibits the shallow-to-deep transition of most clouds within the field. This is consistent with the findings of Chakraborty et al. (2018) that relate shallow convection occurrences with high low-level VWS. On the other hand, the direct VWS effect may generate feedbacks through cloud evaporation or changes in the BL that enable the formation of deeper clouds. This would be consistent with Zhuang et al. (2017) and Helfer and Nuijens (2021) that report on positive correlations between higher low-level VWS and deeper convection occurrences, indicating a non-linear relation. Some of the contrasting results found in the literature may be explained by the balance between the direct and indirect VWS effects. This highlights the need for further observational and high-resolution simulation studies regarding the VWS effect in cumulus cloud fields.

**Code availability**

All codes used in this study are available upon request to the corresponding author (Micael A. Cecchini).

**Data availability**

All datasets used in this study are available upon request to the corresponding author (Micael A. Cecchini).

**Author contribution**

Marco de Bruine designed the reference experiment and provided technical assistance on running the model. Micael A. Cecchini designed the other experiments, performed data analysis, and wrote most of the manuscript. Jordi Vilà-Guerau de Arellano has developed the input conditions for the reference run, supervised the experiments and reviewed the manuscript text. Paulo Artaxo has supervised the development of the manuscript and has reviewed the text.

**Competing interests**

The authors declare that there is no conflict of interest regarding this publication.

**Acknowledgements**

This research was supported by the Office of Science (BER), DOE under Grants DE-SC0021167. This work has been developed within the scope of FAPESP thematic project 2017/17047-0. Micael A. Cecchini was funded by FAPESP grant number 2020/13273-9.

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

**Figures**

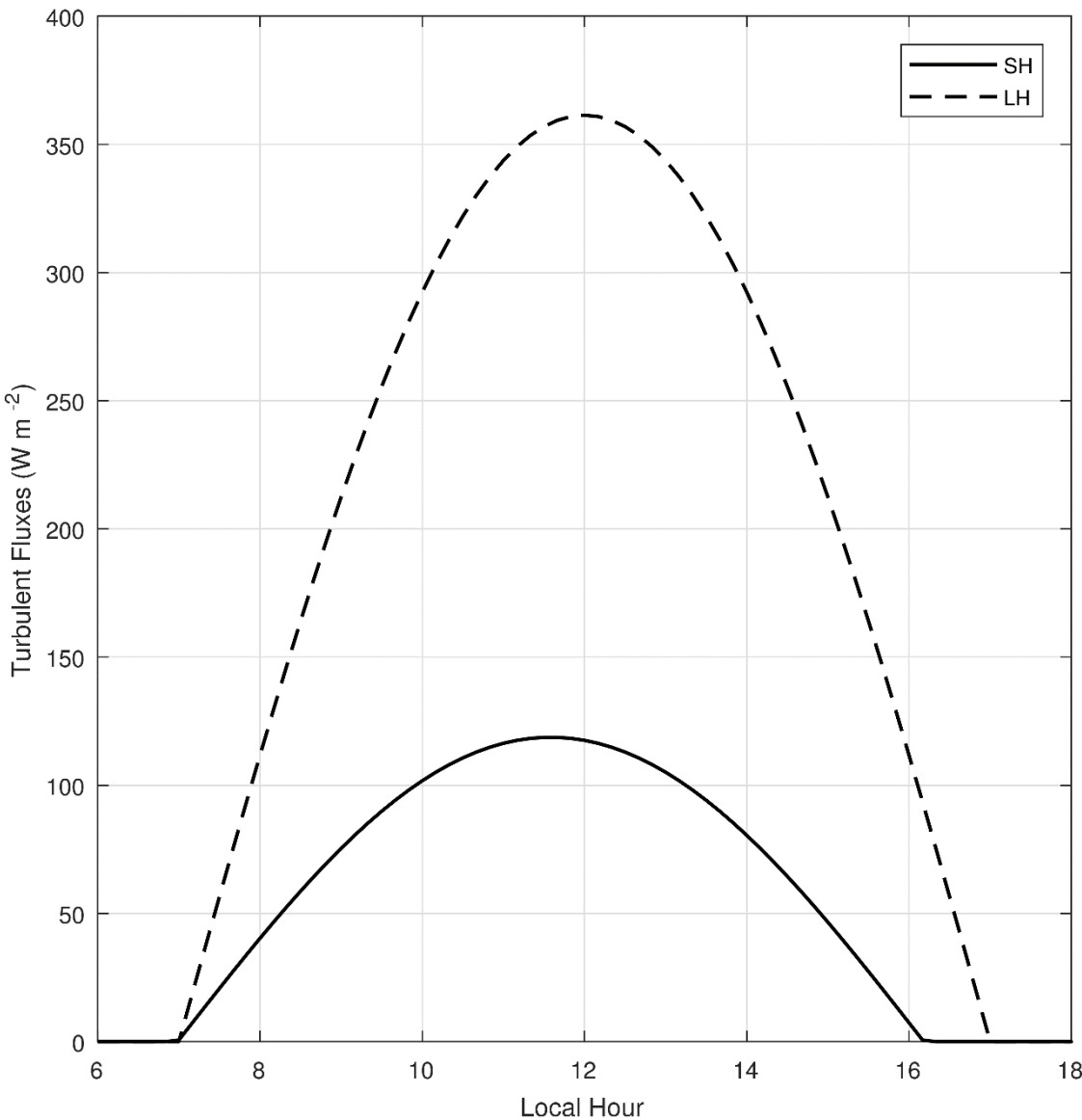

**Figure 1: Prescribed surface turbulent fluxes of all DALES simulations performed in this study. SH and LH are the sensible and latent heat fluxes, respectively. The Bowen ratio SH/LH is mostly between 0.1 and 0.4, consistent with the tropical rainforest environment.**

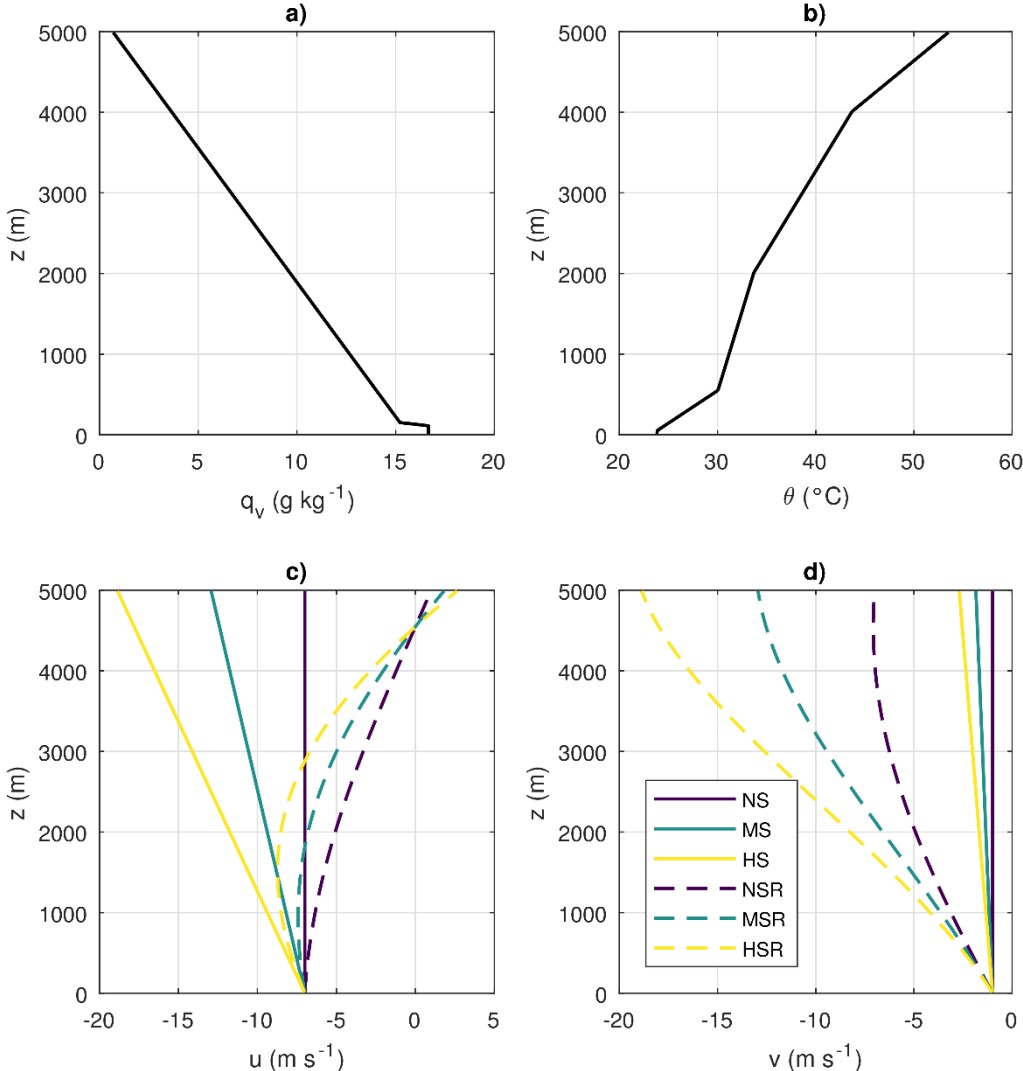

Figure 2: Input profiles of a) water vapour specific humidity $q_v$, b) potential temperature θ, c) zonal wind component $u$ and d) meridional wind component $v$. The large-scale wind forcing is the same as the initial conditions of the respective runs, remaining unchanged throughout the whole simulations. The initial profiles of both $q_v$ and θ are the same for every run, the only difference being the vertical wind variability. Three levels of wind speed shear are defined: no wind speed shear (NS), moderate wind speed shear (MS), and high wind speed shear (HS). A clockwise rotation of 90° in wind direction between 0 km and 5 km is applied in the NSR, MSR, and HSR runs, which have the same wind speed shear as NS, MS, and HS, respectively.

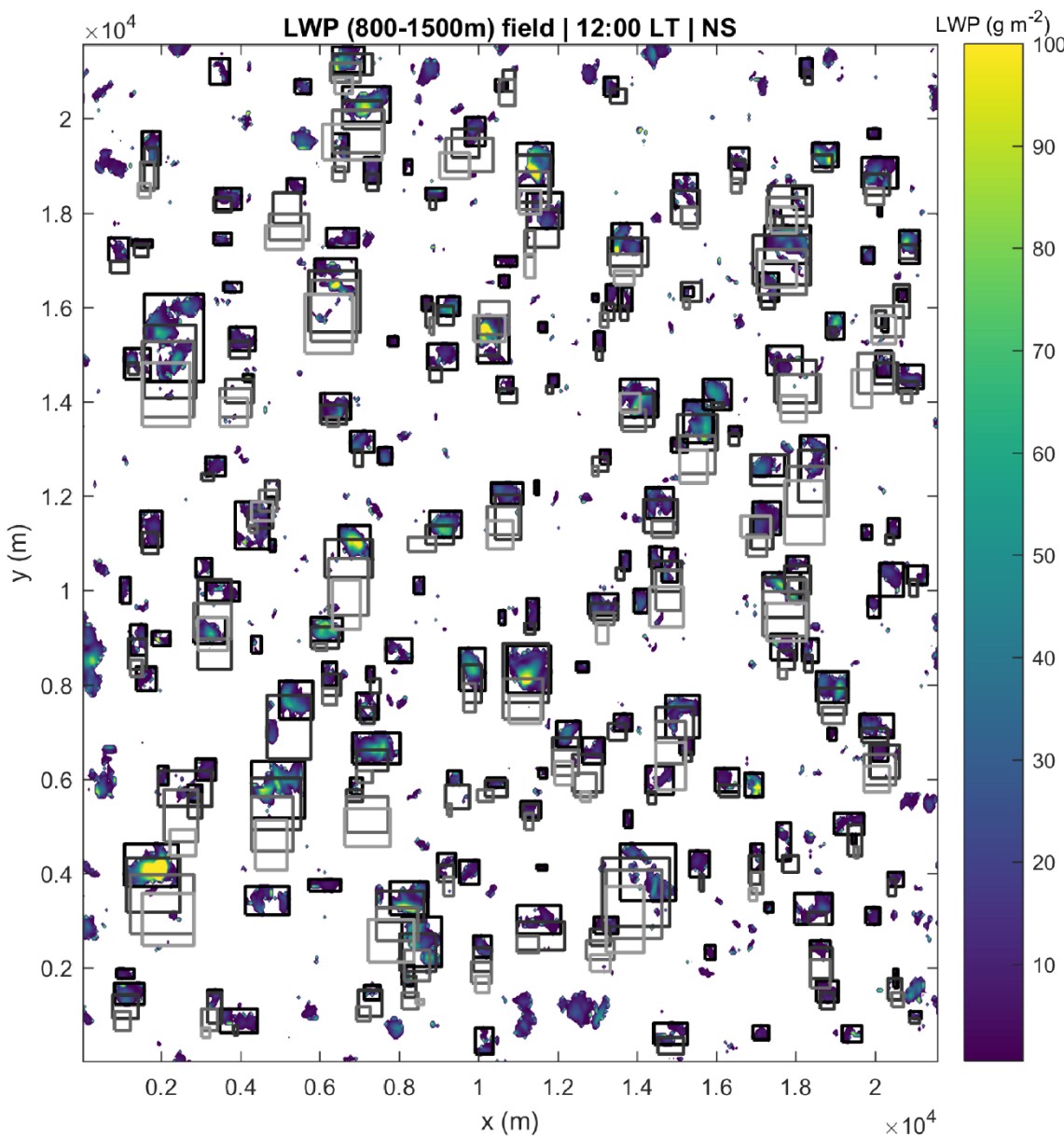

**Figure 3: LWP (between 800 m and 1500 m) field at 12:00 local time for the NS simulation. The tracked clouds are highlighted by rectangles, where black lines represent the current time step and matche the LWP field. Earlier time steps are represented by progressively lighter grey colours, to a maximum of 3 time steps earlier than the LWP field.**

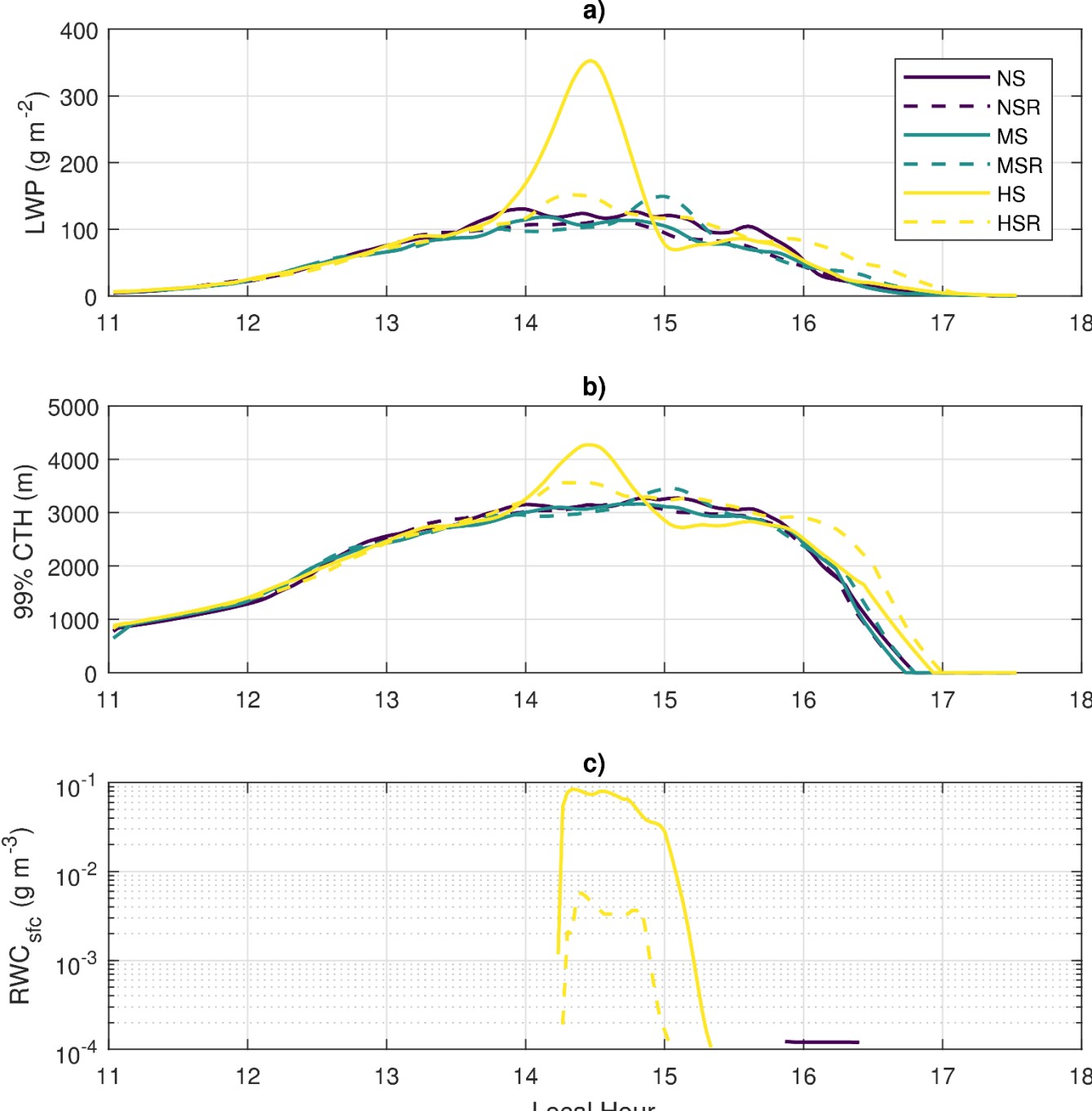

**Figure 4: Time series of domain-wide characteristics: a) averaged liquid water path (LWP), b) 99% percentile of cloud top height (CTH), and c) rainwater content at the surface (RWC<sub>sfc</sub>). The time series have been smoothed by 30-minute moving averages for**
**clarity.**

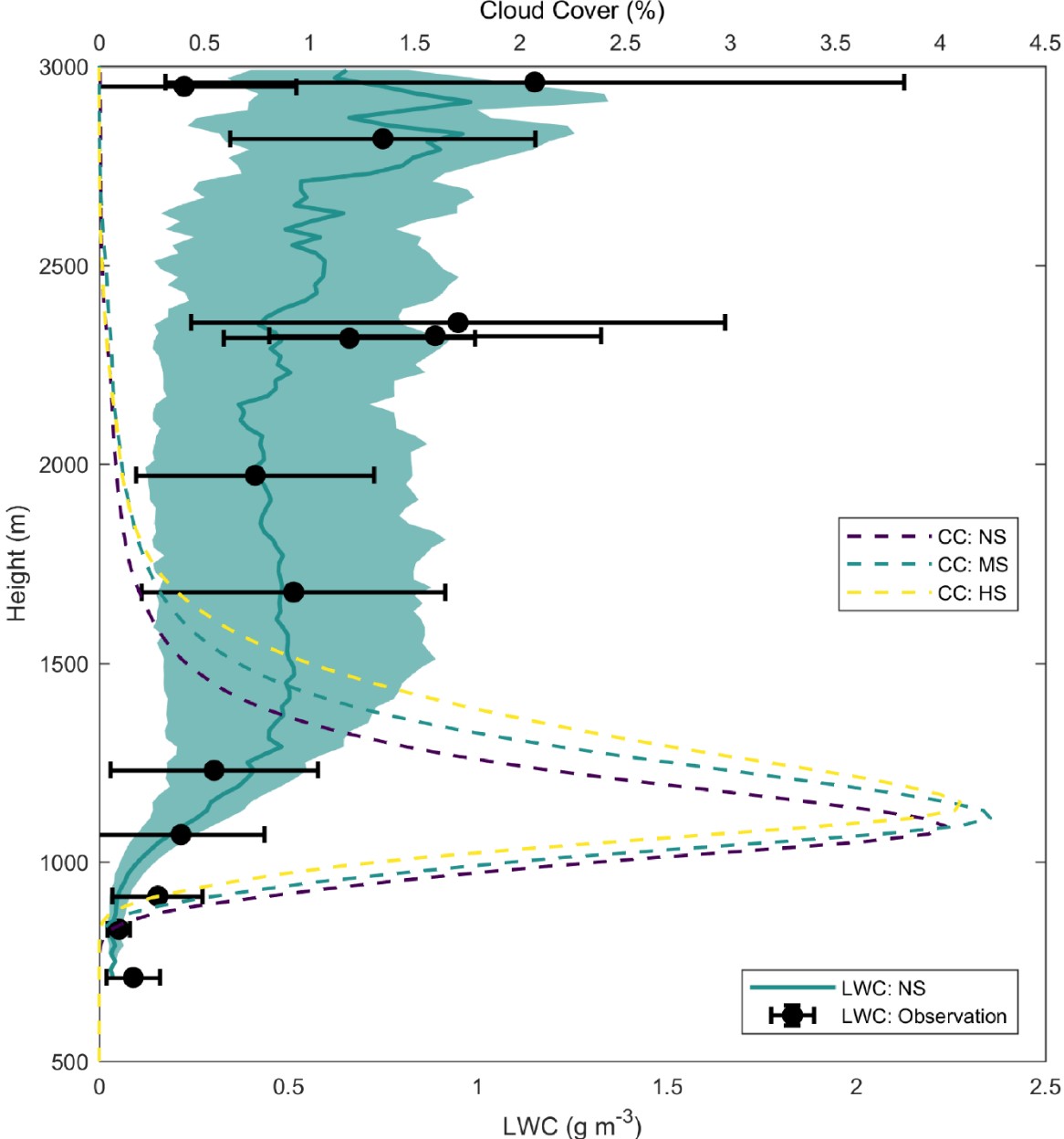

**Figure 5: Vertical profiles of liquid water content (LWC) and cloud cover (CC). The profiles of CC are shown in dashed lines. The profiles of LWC are shown in the green line and shading (model) and in the black data markers and error bars (observations). Observations are taken from flight AC09 during the ACRIDICON-CHUVA campaign in 2014. The model profile of LWC (green line) represents the average of the single-timestep median profiles within the flight period. The green shading was obtained in the same way but represents the average interquartile range in the period instead.**

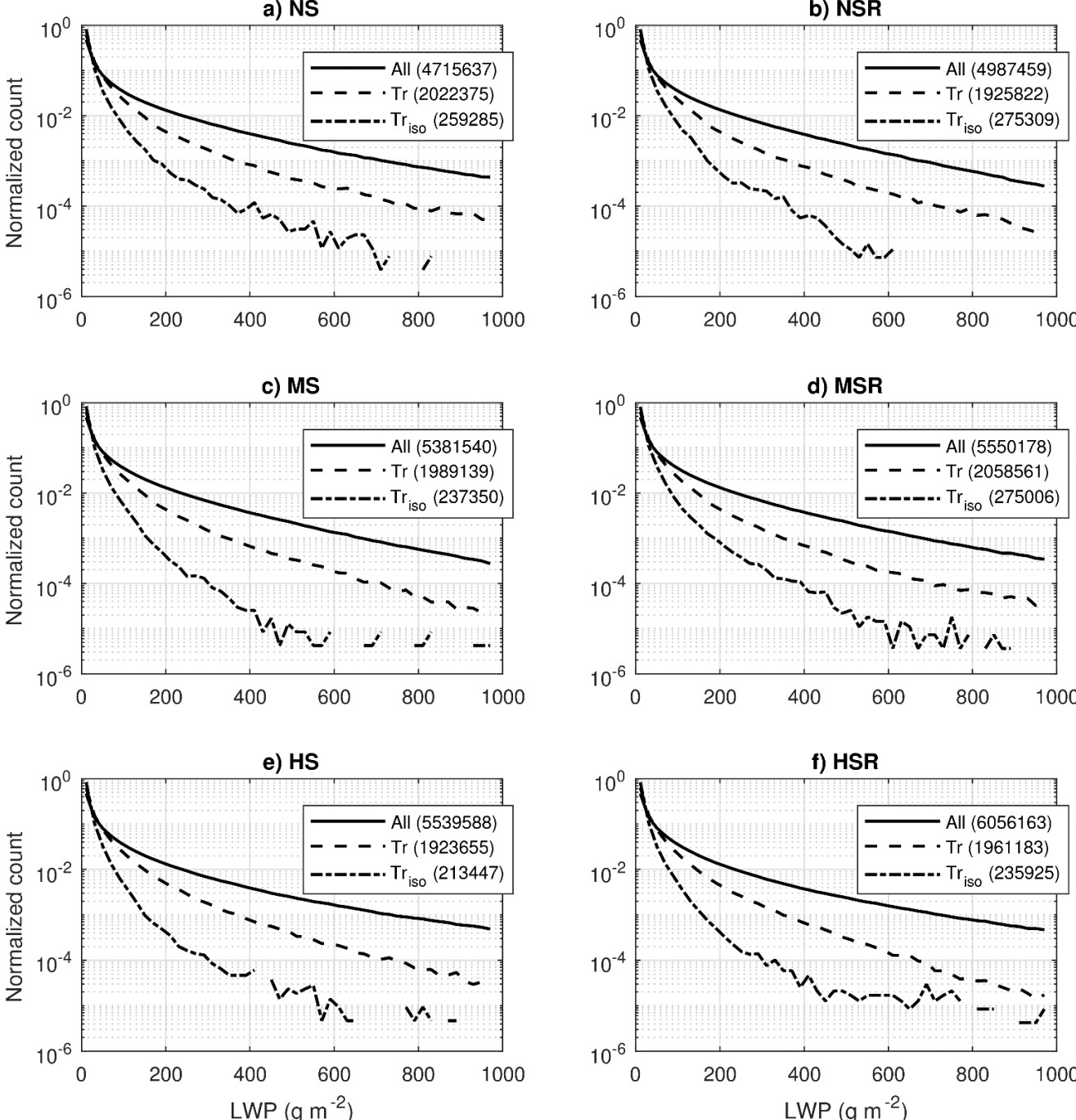

Figure 6: Normalized histograms of LWP for all six runs. The continuous black lines ("All" in the legend) represent the distribution for all pixels in the domain with LWP > 0.6 g m$^{-2}$. The dashed lines represent the distributions only for the pixels that are tracked by ForTraCC ("Tr" in the legend). The dot-dashed lines represent the distributions only for the pixels that are tracked by ForTraCC and for which no merger or split were detected ("Tr$_{iso}$" in the legend). Within the legend, the numbers in parentheses are the total number of pixels in each case. All 2-min output time steps between 11:00 and 18:00 local time are considered.

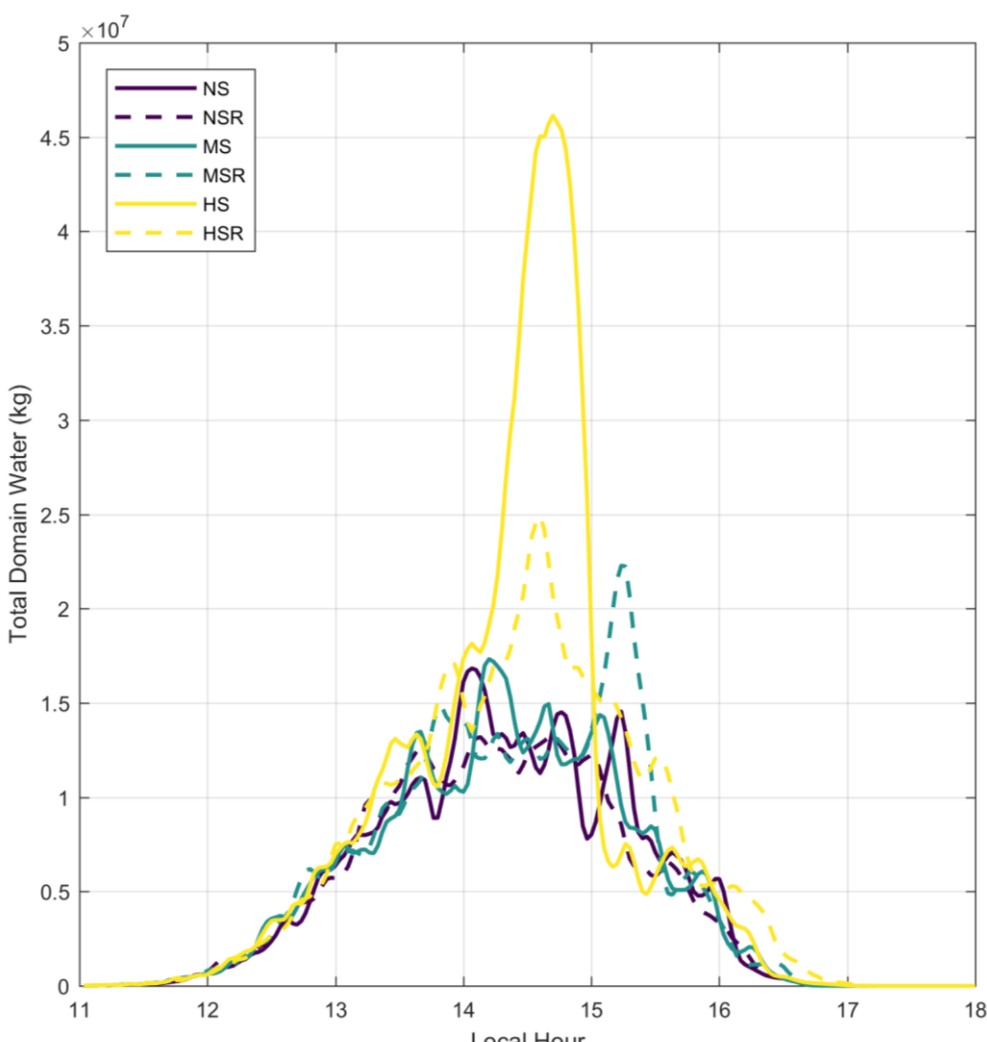

**Figure 7: Time series of the total domain water (kg), calculated as the 3D integration of the cloud and rain liquid water contents over the entire simulation domain.**

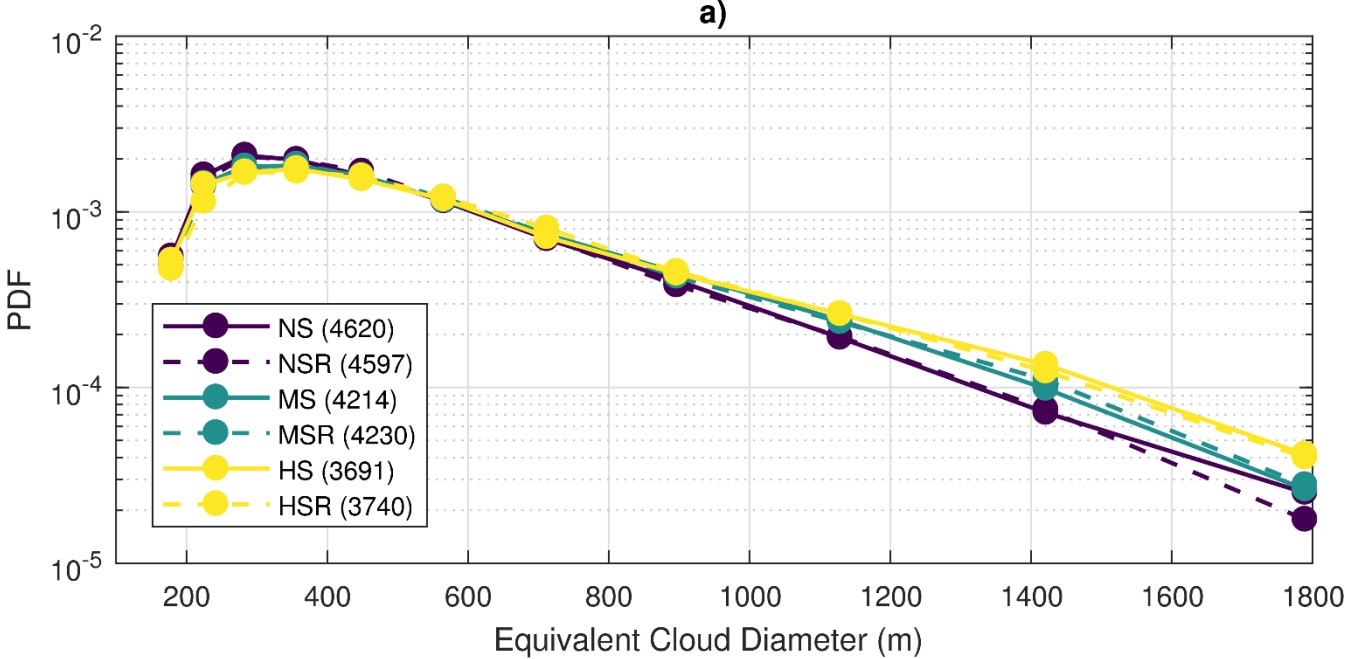

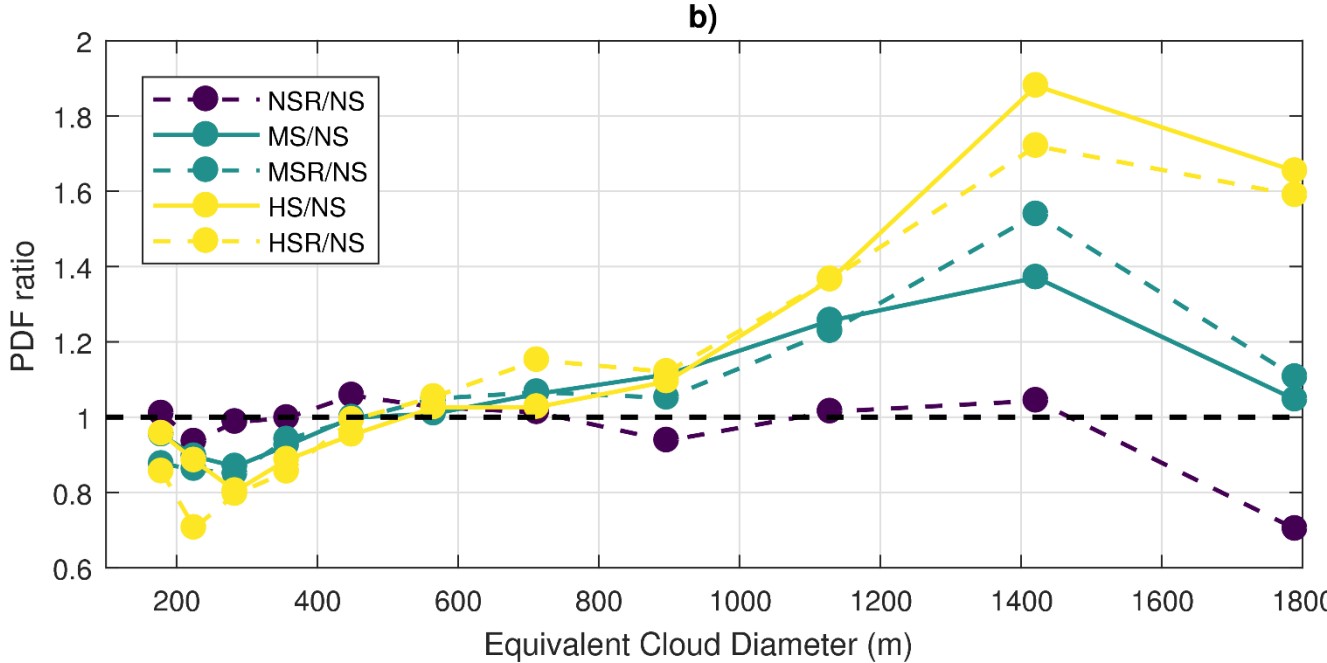

**Figure 8: Probability density function (PDF) of equivalent cloud diameter.** The equivalent diameter is calculated as the diameter of a circle having the same horizontal area as the tracked cloud shape. Panel a) shows the PDFs for all six simulations as a function of the central diameter of every 100-m interval between 100 m and 1800 m. As such, the smallest trackable clouds (180 m) are in the first datapoint. The total number of clouds is shown in parenthesis in the legend of panel a). Panel b) shows the ratio of the PDFs, with the simulation NS as the reference. For statistical robustness, all datapoints represent at least about 50 clouds.

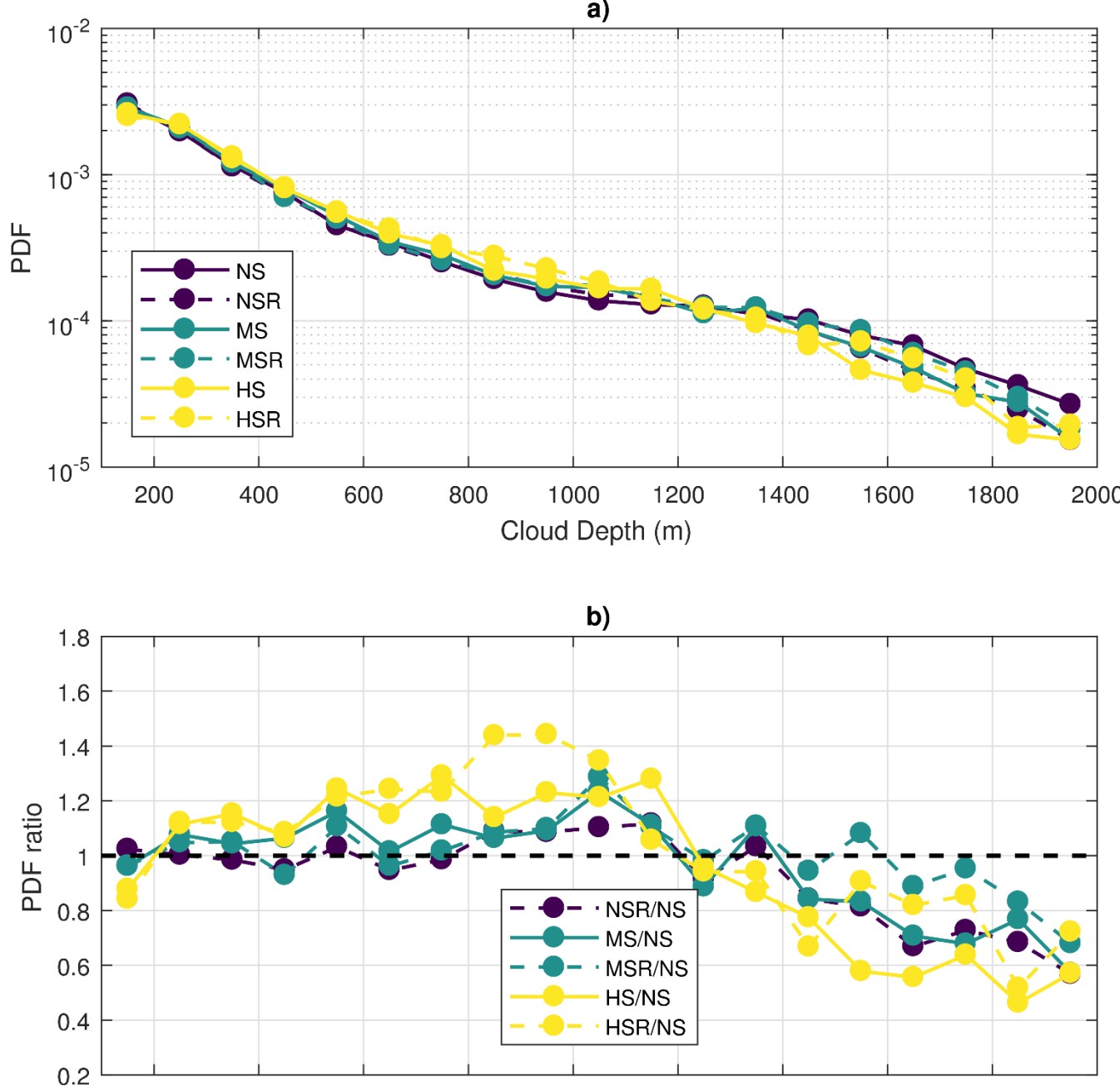

**Figure 9: Same as Figure 6 but for cloud depth. Cloud depth is calculated as the height difference between cloud top and cloud base (using the maximum and minimum height with LWC > 0.01 g m⁻³, respectively).**

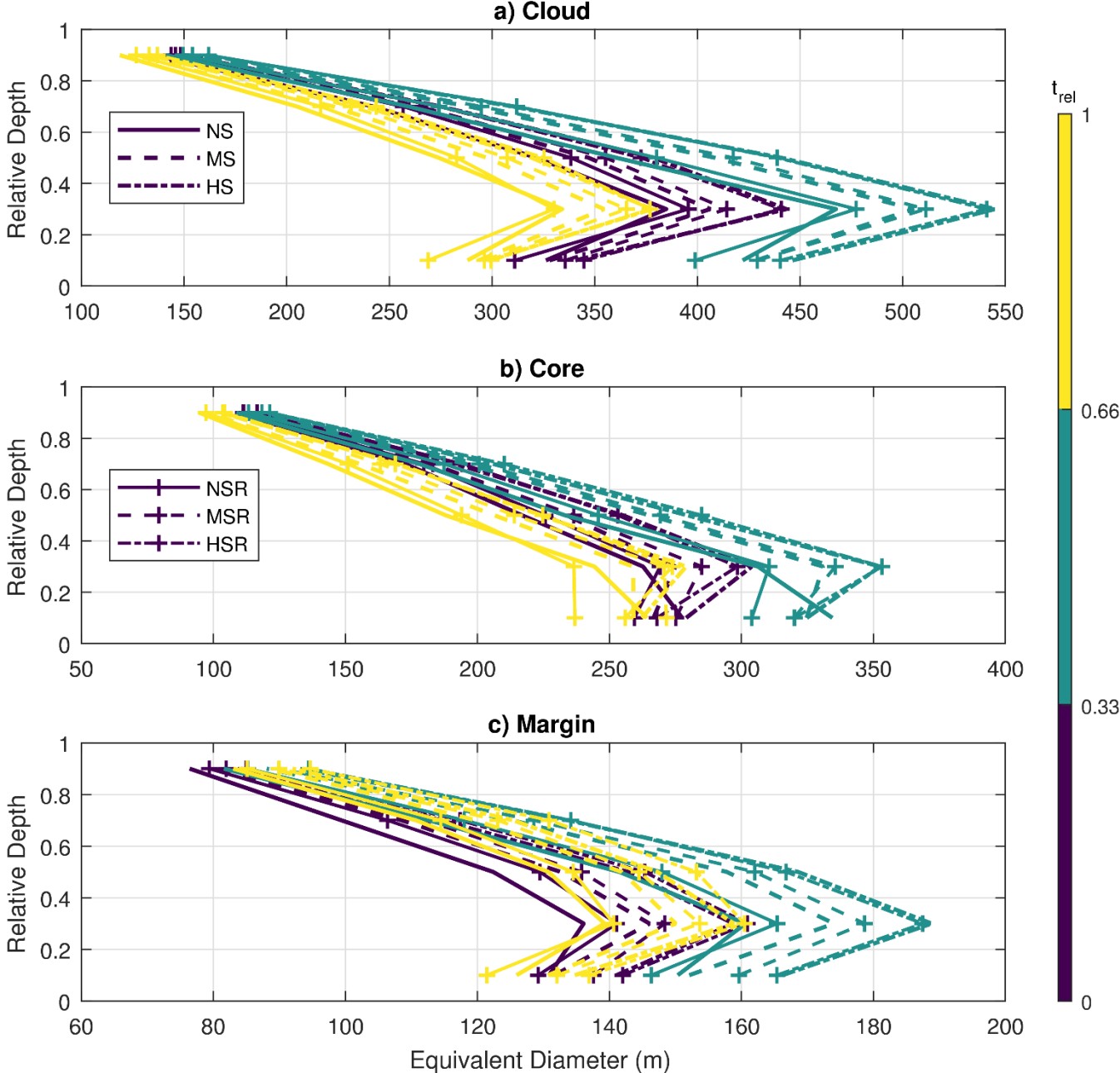

Figure 10: Averaged dimensions of the a) clouds, b) cloud cores, and c) cloud margins as a function of the cloud life cycle and relative depth. The relative depth is a normalized height based on cloud base and cloud top. Values of 0 equate to the cloud base, while 1 represents cloud top. We use five relative depth intervals of 0.2, plotting the datapoints in the centre of them. The cloud life cycle is also normalised ($t_{rel}$, in colours), being 0 at the first initial detection by ForTraCC and 1 at its latest time step. We set 3 life cycle intervals, representing initial, mature, and dissipating stages. The standard deviations are omitted for figure clarity, but they represent approximately 80% of the averages in the depth interval closest to cloud base, decreasing to about 50% close to cloud top.

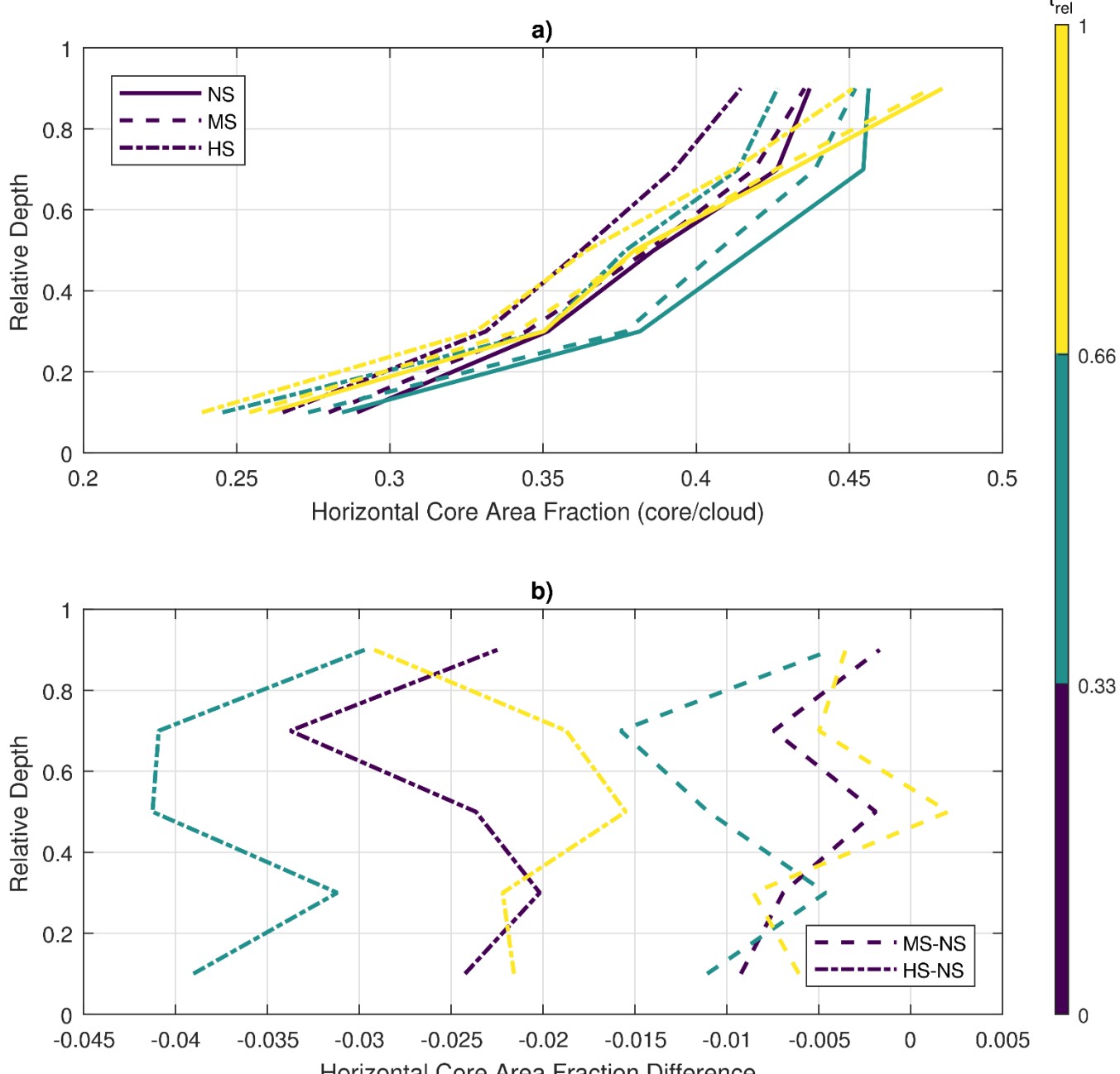

**Figure 11: Similar to Figure 8, but the profiles represent the average core area fraction (i.e., the ratio of the core and cloud areas) in horizontal slices through the clouds. Panel a) shows the actual fraction values, while panel b) shows the differences between the MS/HS runs and NS. The standard deviation of the area fraction represents approximately 70% of the averages close to cloud base, decreasing to 50% close to cloud top.**

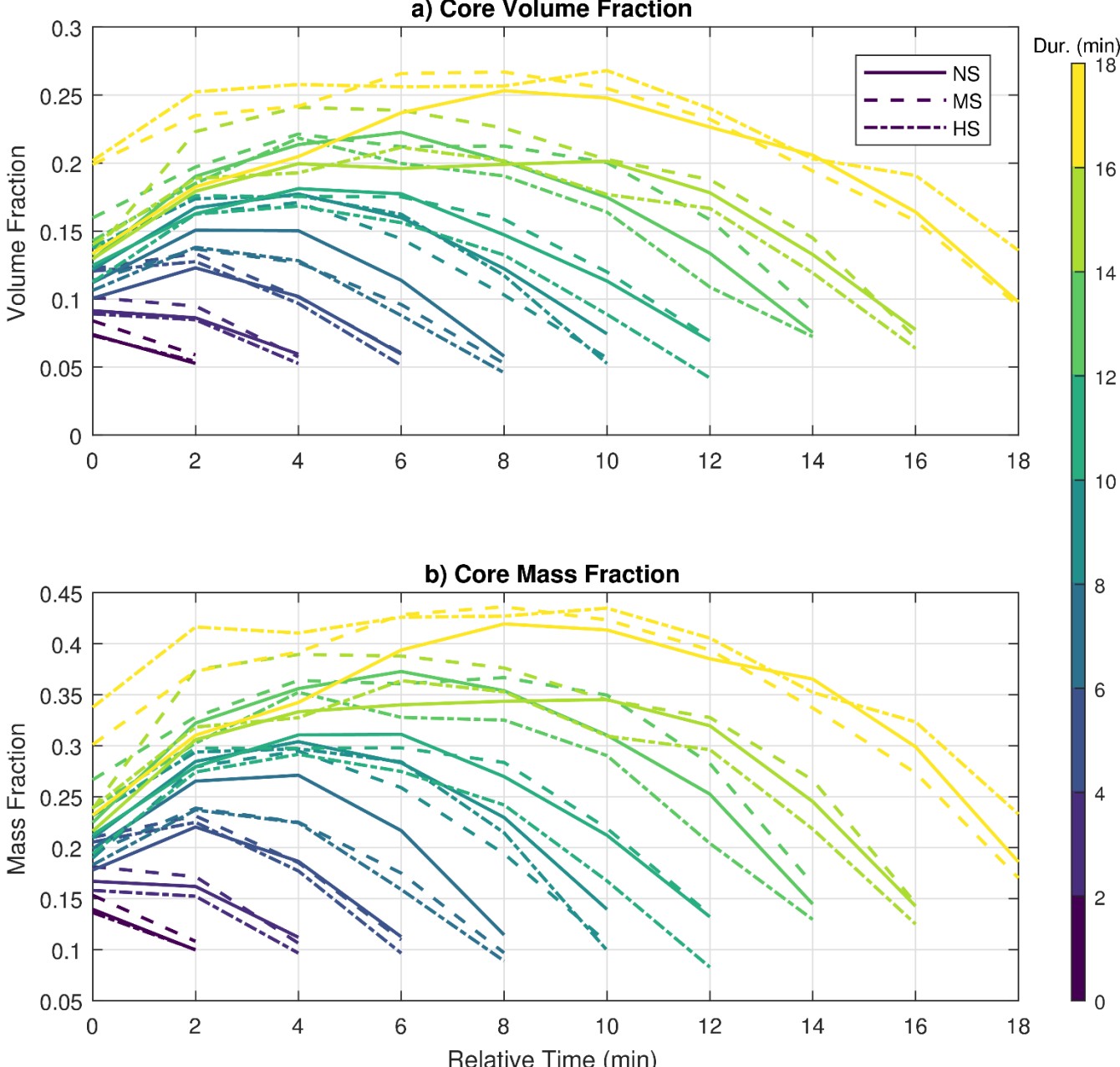

**Figure 12: Cloud core volume and mass fraction as a function of cloud life cycle and total duration. The fractions are obtained by the ratio of the total core volume or mass to the total cloud volume or mass. The total duration is given by the number of time steps representative of each tracked cloud. Clouds only present in two time steps are assumed to have a two-minute duration time, three time steps equating to a four-minute duration and so on. The volume and mass fractions standard deviations are relatively similar for the time steps of clouds sharing the same duration. For the volume fraction, the standard deviations reach up to approximately 0.15. For mass fractions, they have values of up to 0.22 approximately.**

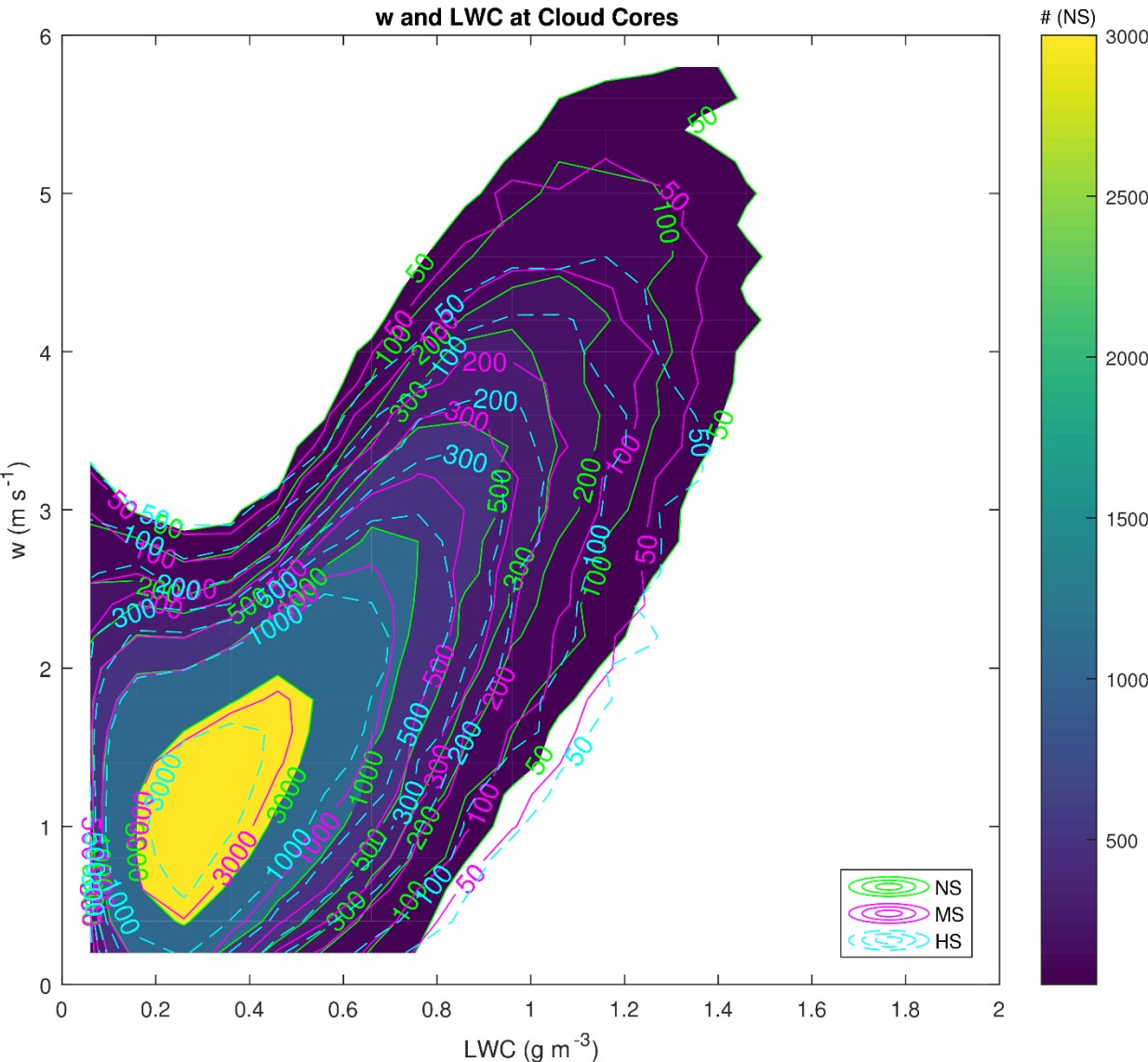

**Figure 13: Distribution of the number of datapoints in the updraughts speed $w$ – liquid water content LWC space for the cloud cores. The colours and the black continuous lines represent the NS run. The MS and HS runs are represented by continuous magenta and dashed cyan lines, respectively. The $w$ interval is between 0.1 m s$^{-1}$ and 5 m s$^{-1}$, with 0.2 m s$^{-1}$ bins. The LWC interval is between 0.01 g m$^{-3}$ and 2 g m$^{-3}$, with 0.1 g m$^{-3}$ bins. The curves in this figure represent the datapoint numbers calculated for those bins.**

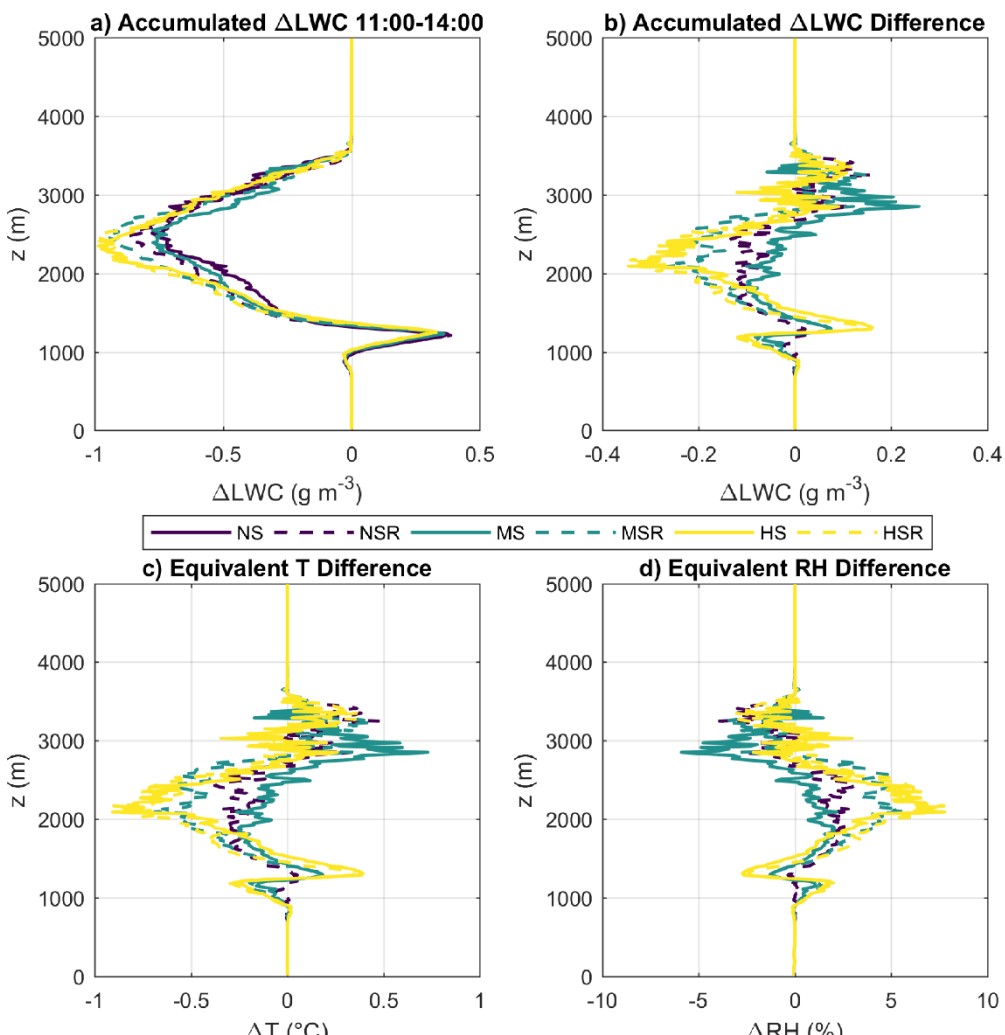

Figure 14: Average effect of the evaporation/condensation rates on the vertical temperature (T) and relative humidity (RH) profiles in the vicinity of clouds. Panel a) shows the time-integrated condensation/evaporation rates between 11:00 and 14:00 local time (i.e., the total water vapour converted to cloud liquid water or the reverse). Panel b) shows the same profiles, but as differences between the runs (having NS as reference). Panels c) and d) show the estimated T and RH differences as calculated from the curves of panel b). This estimate assumes $L_v = 2.5 \times 10^6 \ J \ kg^{-1}$ (latent heat of condensation) and $c_p = 1005 \ J \ kg^{-1} \ K^{-1}$ (specific heat of air at constant pressure).