# Peer review of "Quantifying vertical wind shear effects in shallow cumulus clouds over Amazonia"

_Atmospheric Chemistry and Physics, 2021_

## Referee Comment (RC2)

**Quantifying vertical wind shear effects in shallow cumulus clouds over Amazonia**

By: Micael Amore Cecchini, Marco de Bruine, Jordi Vilà-Guerau de Arellano, and Paulo Artaxo

**General comments:**

This manuscript studies the evolution of shallow cumulus clouds over the Amazon in response to vertical wind shear. The role of wind shear is not well-studied and this study can make an important contribution to the field. The results indicate that a high value of vertical wind shear leads to (marginally) larger and shallower clouds, with a relatively smaller core area. They also find that there is more (localized) evaporation in the cloud layer under shear which may destabilize the cloud layer and have consequences for subsequent convection. Similar to what previous studies have found, the results also suggest there is not a linear relationship between cloudiness and wind shear, whereby high vertical wind shear can also increase the total water content present in the domain.

I think the authors did a nice job at explaining the cloud tracking algorithm that they use, at synthesizing the results and reflecting on the findings in light of the uncertainties in the simulation, the tracking and the complexity of the problem. Yet I agree that there is important information missing regarding the setup of the winds that requires clarification, which I outline below. There are also aspects of the sampling of core and margins that are unclear and I believe there are opportunities for using the cloud tracking statistics to explain how clouds of different sizes and LWC contribute to the overall increase in total water in the domain.

I would advise a **major revision** based on the **main comments** below:

1. The manuscript does not describe the evolution of wind profiles, the large-scale wind forcing and the surface stress. One aspect of the setup that is not clarified is that the winds will evolve during the course of the simulation: they will be mixed and slowed down throughout the BL in response to surface stress and the winds will gain an ageostrophic component. The shear present at noon would be different from the initial shear profiles.
   The development of the wind profiles during the simulation influence how we interpret the differences in clouds that are presented as a result of differences in shear in the cloud layer.
   A directional shear is prescribed in the initial profiles that turns counter-clockwise with height: usually you have a veering of the wind with height e.g. a clockwise wind turning (while friction would lead to a backing of the wind and a counter-clockwise wind turning towards the surface). Your geostrophic winds at 5 km imply that upon friction near the surface (or turbulence in the boundary layer) a westerly wind component will develop. Could this decrease the shear throughout the simulation period and lead to smaller shear differences as the simulation evolves?

   Shear in the mixed layer can trigger indirect (as you mention, non-linear) effects. Helfer and Nuijens (2021) find that under forward shear in the subcloud layer there is the tendency for convection to become more organized and deepen more (although vertical updrafts within clouds are still hampered due to the slanting of clouds).
   The positioning of clouds relative to their sub-cloud layer roots can play a role in convective deepening by promoting stronger updrafts already in the subcloud layer. Under forward shear in the sub-cloud layer the updraft and downdraft region become

more separated and the coherent circulations and resulting convergence may be strengthened.

https://agupubs.onlinelibrary.wiley.com/doi/full/10.1002/2015JD023253

The different wind speeds that develop at different heights is also relevant for the numerical diffusion of clouds, which is an issue with LES and the use of an all-or-nothing cloud scheme. The differences in cloud diameters between the shear cases is only 1 or two grid cells, something which I can imagine would be influenced by the differences in wind speed in the cloud layer (and such differences will be there even though a Galilean transform is applied, because the wind profiles are sheared).

2. Can the authors clarify the core and margin selection and the results in Figure 8: am I right that the diameter of the core and margin sum up to be larger than the cloud diameter? This means that margins are occupying an area that is roughly the cloud diameter - core diameter (not really the margin or edge), or even larger, which would imply that there is overlap in the classification. It was my understanding that you try to contrast edges versus cores of the cloud, how should I be interpreting this?

3. It would be nice if the authors make (more) use of tracking algorithm to delineate the contribution of clouds sizes/depths to LWC/LWP under different shear and to indicate whether there is more organization, which in studies of deep convection is clearly related to the wind shear imposed. For instance, are there more merging events?

   In section 3.2 you argue that whole-domain properties such as the total domain water are heavily influenced by infrequent and large clouds. I don't think you actually show this (yet). You could show this by plotting the cumulative cloud liquid water as a function of the cloud equivalent diameter (using all individual clouds). I'm not so sure whether it is just a single deep cloud that is responsible for the larger domain total water under high shear, because you also observe that under HS there are more frequent clouds with (intermediate) cloud depths of 600 - 1200 m: could they cumulatively explain a larger total water?

   And can you show whether the larger domain water content under shear is contributed by the larger core dimensions in absolute term, or the larger margin dimensions in absolute terms?

   In section 3.2 you also discuss that there is more evaporation under higher shear because clouds have a larger area due to tilting. At the bottom of page 13 you argue that there is more evaporation because of more liquid water content in the atmosphere (Figure 5). The cores are relatively small, but there is more cloud overall. Related to the above, it is not clear which clouds - and whether it is their cores or their margins - contribute most to the overall LWC in the domain. Can this be shown?

4. With respect to the results that are in Figure 12: Figure 4 showed that under higher VWS there is a larger difference between all cloudy pixels and tracked pixels, also at larger LWP's. There must thus be a lot of not-considered cloud pixels in the simulation that also evaporate and contribute to the heating and moistening tendencies.

   Have you compared the evaporation differences under shear to the domain-mean tendencies of humidity /temperature? All simulations have the same surface moisture

/heat flux, but the humidity and temperature may be distributed differently. Do the differences in Fig 12 carry over to the total domain-mean tendencies or do the cloudy pixels not considered in the tracking also play a role?

The difference between localized (near cloud) changes in evaporation and domain-mean humidity and temperature tendencies is unclear in the results, but better discussed in the summary/ discussion.

Some more minor comments are these:

- A rigorous grammar & spelling check would be needed before publication.
- Section 2.3: I would say that these days 21 x 21 km^2 is not that large. For shallow convection the typical domain size is now at least 50 going up to 125 km in one direction to allow for mesoscale dynamics to take place.
- P7L213: About the 10 km radius: At 7 m/s this corresponds to about 23 min, so that is the maximum length to be tracked .. Seems ok with respect to a 20 min life cycle of small clouds? (I guess you will present lifetime statistics later, but would fit here as well).
- Section 2.3.1: L 224: "we exclude …" Do you repeat the process by excluding the larger shape from the 3D volume? (is that smaller shape tracked on its own and kept in the statistics despite being in close proximity to a larger clouds at some point?) I am asking because of the difference between isolated clouds and all tracked clouds in Figure 4. How much of that is because of excluding splitting or merging cells versus removing cells in close proximity or removing stacked clouds?
    - I am asking because of the difference between isolated clouds and all tracked clouds in Figure 4. How much of that is because of excluding splitting or merging cells versus removing cells in close proximity or removing stacked clouds?
- Would section 3.1 be better placed in section 2 along with the description of the algorithm?
- Figure 8: It is confusing that the colours which were used before to indicate the amount of shear now denote the relative time duration
- I found that Figure 10 received quite some discussion but did not really aid the rest of the story.
- Figure 11 is quite busy and complex. Could it be changed to a conditional PDF (conditioned on bins of LWC) that show the distributions of vertical velocity for the NS, MS, HS cases as just a line plot? Or some other version that more quickly allows one to see the main result. BTW: does it seem that the differences in vertical velocity/updrafts are mainly pronounced at larger LWC?
- P13 393: "VWS not only reduces the dimensions of the core": you miss denoting "relative" I think, because Figure 8b shows larger dimensions for the cores under shear.
- P15 last (bottom) paragraph: What you write here could have also been addressed with "conventional" cloud (core) sampling in LES. Can you better emphasize the value of the tracking here? As mentioned, I think what can be really valuable is quantifying what are the cloud (sizes) that are most affected by shear: you show that while the small clouds are dissipated more quickly, the intermediate clouds may benefit and get larger. While clouds are overall shallower under more shear, there are a few much deeper clouds that develop under shear.
- P16 512: "by changing the conditions of cloud formation": I think this can and should be explained a little further…. what conditions?

---

## Author Comment (AC2)

**Reviewer #1**

**Overall comments**

The revised manuscript is in good shape. I wonder if the authors could add more analysis for the effect of the directional shear, since this is one of the main findings from this study. Does directional shear tilt clouds as much as without it? I do not feel this analysis is not necessary but including it makes the manuscript widely read. After the following minor corrections, I think the manuscript is ready for publication.

**Response**

We are thankful for the pertinent comments from Reviewer #1 and for coming back for the second round of review. We agree that a deeper analysis of the directional shear would be interesting. However, we found it hard to highlight the directional shear results based on what we wanted to analyze in this manuscript. Our primary goal is to quantify the effects of vertical wind shear on the physical dimensions of the clouds and on their core/margins properties. Our results show that the directional shear did not present significantly different results than the wind speed shear, which is why we do not focus much on the directional shear. There is sure to be other interesting aspects of directional shear to look into, but for the purposes of this manuscript we did not feel the need to highlight it as much.

**Minor Comments**

*Responses are italicized and in red.*

- line 24: Delete "(w)" and "(LWC)". *Done.*
- line 46: Since "ACRIDICON-CHUVA" is highly likely an acronym and its first appearance, it should be expanded. Check ACP's formatting rule for acronyms. *Done. Expanded the GoAmazon2014/5 name as well.*
- line 52: Define "VOCs" and "UT". *Done.*
- line 56: "...other sources of aerosol particles." I wonder the discussion (from line 45) for this topic is necessary? It is certainly interesting, though. The discussion could be made simpler. *We have removed two sentences to reduce the discussion of this topic. We consider the topic itself to be important because it is one of the motivations behind the study of VWS effects in cumulus clouds – i.e., whether VWS affects or not their capability of growing into deep convective clouds. Our results suggest that VWS may indeed help the formation of deeper clouds.*
- line 61: Define "VWS" here because of its first appearance in the text. *Done.*
- line 107: With what time stepping scheme? *The time stepping is adaptive, with a maximum of 1 second (mentioned in Section 2.2).*
- line 108: "radiation" => "radiative". *Done.*
- line 121 and below: Use italic face for variables such as "dS", "T", etc. in both text and captions. *Done.*

- Equation (2): The round bracket in front of the dS_macro in the right hand side is not closed yet. *Good eye. Closed it.*
- line 126: "The supersaturation sink coefficient (gamma)" should be introduced immediately after its first usages in Equation (1). *Done.*
- line 163: I still do not know how u and v wind profiles are maintained over the simulations. *We have clarified that the wind speed/direction increments are applied to the initial and large-scale forcings.*
- line 199: The moving domain with the horizontal wind speed at 1 km height should be mentioned in the experimental setup. This treatment is also beneficial to reduce numerical diffusion. *We appreciate the suggestion but feel like this information is a good link to discuss the tracking algorithm. In that sense, we would prefer to keep it in Section 2.3.1.*
- line 252: Is the cloud top over 4 km (fig. 4) too close to the domain top of 5 km? Which level is the damping/sponge layer applied? The configuration should be mentioned in the experimental setup. *The dumping layer starts at 4 km, so indeed the cloud top is within it. We now mention it in the experiment description as well as in this first paragraph of Section 3.1.*
- line 287: Should "the statistics of the ... bias towards smaller clouds..." be "... bias towards larger clouds..."? For smaller clouds, statistics well match up. *The meaning here is that the statistics of the tracked clouds will bias towards smaller ones because they are more numerous. The largest clouds are not tracked since they are way more likely to have pixels crossing over the boundaries. The domain-wide statistics are indeed biased towards the largest clouds, however.*
- line 411: "continuous lines" should be "contour lines" Correct in text as well as captions. *We use "continuous lines" here to distinguish from the dashed lines present in the same figure.*
- line 426: Should "...latent release..." be "...latent heat release..."? *Absolutely, thank you.*
- Figure 1: "Prescribed turbulent fluxes..." => "Prescribed surface turbulent fluxes...". *Done.*

**Reviewer #2**

**Overall comments**

I thank the authors for their revised version and explanation of the changes made. In their revision, they have addressed the comments of reviewer 1 that concerned updating the tracking algorithm to ensure a larger statistical sample and providing more evidence of general results such as cloud profiles and time series of LWP/precip. This has surely improved the presentation of the simulation case itself. With respect to my comments, admittedly the authors have not implemented many changes and dismissed the opportunity to derive some more conclusions from the cloud tracking to address the influence of cloud sizes on the evolution of the boundary layer and its distribution of

water. I accept their decision. However, I would ask the authors to still address the concerns regarding the evolution of wind and specification of wind forcing (which are also raised by reviewer 1), which they have not address appropriately.

**Response**

We are thankful for the insightful comments from Reviewer #2 and for the continued efforts in this second round of revisions. The comments from Reviewer #2 are very relevant and we detail our responses below.

**Major Comments**

**(Major Comment #1)**

In the first review I asked whether the authors can clarify how the wind speed profiles evolve during the simulation, because this influences how we interpret the differences in clouds that are presented as a result of differences in shear in the cloud layer. The authors have shown in their response how the profiles change within the boundary layer, and comment that the HS(R) simulations develop about 1 m/s more wind in the boundary layer, which would lead to a more rapid development of the boundary layer – they also included this statement in section 2.2. I appreciate the authors showing me these profiles and I would be fine if the profiles themselves are not added due to the (already large number of) figures. However, the consequences and description of what they show should be correct and I don't think this is the case yet: First: the additional 1 m/s cannot lead to a more rapid BL development, because the surface fluxes are prescribed: then how (e.g., through a few deeper clouds), but are you sure this is significant? Looking at the boundary layer that has developed in the simulations (in the wind speed profiles in the reply to reviewers), they boundary layer height is approximately the same. More importantly though: the issue is not so much whether the wind speed is different, but what differences in shear develop. As the figures clearly show, at midday and in the afternoon, most shear is concentrated in the surface layer (up to 200 m) and at the inversion, while in the well-mixed boundary layer, between 200 – 1200 m, there is little shear in the HS(R) simulation. This is where the majority of your cloud fraction is sitting. In other words, while the simulations initially differ much in the amount of shear prescribed, as the boundary layer develops the amount of shear in a large part of the cloud layer has already been mixed away. I think this requires more thought or clarification in the text, or even a discussion! especially because the evolution of LWP and cloud top height are overall rather similar across the simulations and seemingly independent of the shear. It should be discussed that this might be in part because the shear is (already, quickly) mixed away.

• The authors have not sufficiently clarified the large-scale geostrophic wind forcing that is applied – I note this was also requested in major comment 1 of reviewer 1, and seemingly not addressed / edited in the text. Perhaps I missed it, but I don't see any

description of large-scale wind forcing. Figure 2 only reads input profiles, which would mean initial profiles, it does not write large-scale wind profiles anywhere.

• NOTE: The profiles of the wind direction show that you apply a clockwise turning of the wind with height (wind veering from ~ 100 – 190 deg): the text still says that you have a counter-clockwise turning.

• The authors have not addressed the concern that it needs to be specified how the Galilean transform is done, because the differences in cloud diameters between the shear cases is only 1 or two grid cells, something which could be easily influenced by the differences in wind speed in the cloud layer (and such differences will be there even though a Galilean transform is applied, because the wind profiles are sheared).

**(Response to Major Comment #1)**

We thank Reviewer #2 for this comment, it helped us have more clarity on the processes we are discussing in this manuscript. For clarity in our response, we will split it into topics, as follows:

- The extra 1 m s$^{-1}$ wind speed in the HS(R) runs and the differences in boundary layer (BL) height

We believe we may be discussing two different things at the same time here. In one hand, we have the added ~1 m s$^{-1}$ of wind speed in the BL in the HS(R) runs as compared to NS. This was just an observation made to the Reviewers in the response material and is not discussed much in the manuscript aside from the comment in Section 2.2. On the other hand, we have the addition of wind sped shear throughout the whole vertical domain, which naturally also adds wind shear within the BL itself. We do not claim that a 1 m s$^{-1}$ wind speed addition by itself causes the growth of the boundary layer. Instead, we mention in Section 4 that this is likely related to the wind shear. We discuss this within the context of the Henkes et al. (2021) work because it is a nice observational reference for the same region. We have added comments to Section 4 to clarify this point, with the following reasoning: with added wind speed shear, the entrainment rates at the top of the BL will be increased (Pino et al., 2003), which causes it to grow faster. This is consistent with the Henkes et al. (2021) findings that the low-level jet is stronger on days with deep convection – i.e., in that case, the increase in turbulence within the BL would be due to stronger low-level jet. In our case, the results capture such mechanism in a different way because our wind shears are linear. Nonetheless, our sheared runs have added wind shear within the BL, as well as in the entrainment layer, which produces the extra BL growth even though the surface fluxes are prescribed.

To make sure our BLs in the HS(R) are indeed higher than in NS, we have calculated the BL height using the bulk Richardson method with a threshold of 0.25. We show the results in the figure below. On the upper panel, we show the time series of BL height for the NS(R) and HS(R) runs, while the bottom panel shows the differences. This figure shows that prior to the deepest cloud at 14:00 in the HS run there is a

~150 m difference between the BL height in HS and NS. This is consistent with the mechanism we described earlier where the added wind shear induces more entrainment and accelerates BL growth. We have added a comment on this point in Section 4 to improve the discussion of this important topic brought forth by the Reviewer.

[Figure]

-   The wind shear mixing in the layer between 200 m and 1200 m

We agree with the Reviewer that the wind shear is low between 200 m and 1200 m. This is due to the mixing within the BL, but it should not affect clouds as much. We note that the BL height is usually close to cloud base: in our calculations using the bulk Richardson number approach with 0.25 threshold, the cloud base height is within ~150 m of the BL height. Of course, this would change depending on the threshold used or even if other methods to estimate BL height are used. Nevertheless, the clouds should develop mostly above the BL, where the wind shear is stronger. For example, at 12:00 local time, the figure above shows that the BL height is approximately 1000 m. From our previous response to the reviewers (vertical profiles of wind speed), this level is where the wind

shear starts growing again. Noting that our clouds are at least ~100 m deep (Figure 9 in the manuscript), they should indeed develop under wind shear conditions and not under the mixed wind regime as mentioned by the Reviewer. Note also that in Figure 5 of the manuscript the maximum cloud cover is slightly above 1000 m, which should be close to cloud base height – this also indirectly indicates a deeper BL in the HS run because of the higher height of cloud cover maximum.

- Mention of large-scale wind forcing

We now mention the large-scale forcing in Section 2.2 and in the caption of Figure 2. The large-scale winds are the same as the input wind profiles and are kept constant throughout the simulations. We thank the Reviewer for noting this lack of information in the manuscript.

- Clockwise vs counterclockwise wind rotation with height

The Reviewer is correct, the winds have a clockwise rotation and not a counterclockwise rotation as we stated in the manuscript. Thank you for the correction.

- The Galilean transform

The Reviewer raises a good question – i.e., whether the different wind speeds mentioned in Section 2.2. would affect the cloud sizing. Here we note that we are using a tracking algorithm that follows the clouds regardless of their displacement velocity. Therefore, the Galilean transform is implicitly taken care of by the tracking algorithm and the cloud size calculations are independent of their displacement velocity.

**(Major Comment #2)**

In my first review I also asked the authors to clarify the core and margin selection and why the diameter of the core and margin sum up to be larger than the cloud diameter? The authors reply: "the reason is that the data sampling is slightly different for the cloud, core and margins" and explain how it can happen that some pixels will not be classified at all. But that would imply that the core and margin would sum up to be less than the cloud diameter, and my concern is that sometimes they add up to be more than the cloud. The answer is not entirely satisfying. The other answer provided - that different shapes of the cloud, core and margins play a role here - would be more satisfactory, and if the authors are really certain that this is the case, they should state it in the text. It would have been nice if the authors double checked that the sampling works as they expect.

**(Response to Major Comment #2)**

We have given this issue much more thought and went back into the data to analyze in detail. What we find is that, indeed, the sum of the core and margins lengths is almost always slightly larger than the cloud dimension. Upon further inspection, we found it

hard to confirm the effects of different shapes – i.e., the response that would be more satisfactory to the Reviewer. Given the large number of clouds, it is quite hard to quantify the effects without a complex algorithm that detects the shapes and then compares them between the cloud, cores and margins.

What we can confirm, however, is that there are indeed some sampling differences between the cloud, core and margins pixels used in the calculations shown in Figure 10. Note that the core and margins classifications are quite restrictive because they require all pixels to have all positive (or negative for the margins) updraft speed, buoyancy and supersaturation. So, at a given height in a tracked cloud, it is possible to have pixels classified as "cloud", but none classified as "core" or "margins". Or we can have situations where there is only "cloud" and "core" at a given height, for instance (and the same for "cloud" and "margin"). In the end, this produces a non-trivial comparison between the cloud, core and margins lengths because they have different samples. Overall, we found that the cross sections that are fully classified as "cloud" (with no "core" or "margins") usually have smaller dimensions than cross sections that contain either "core" or "margin" classifications. Therefore, the cloud dimensions in Figure 10 have a bias towards smaller sizes and the opposite happens with the core and margins dimensions. We have added this explanation to the text.

---

## Author Response (AR1)

**Reviewer #1**

**Overall comments**

This manuscript attempts to reveal a complex effects of vertical wind shear on shallow cumulus clouds by applying an off-line cloud tracking model to large eddy simulation data. Large eddy simulations are configured to simulate shallow cumulus clouds over Amazon.

The Lagrangian analysis has an advantage to provide statistical understanding of time evolution of clouds, i.e., lifecycle of clouds. The authors are able to present the lifecycle of an average cloud under non-shear and shear environment. The effects of vertical wind shear to the lifecycle of the simulated clouds are consistent with previous studies.

Although I think the results obtained with their Lagrangian analysis are new for this topic, the manuscript lacks a backup discussion and supportive figures. Also, the results are qualitative rather than quantitative contrary to their intention to provide quantitative arguments. There are several caveats that have to be cleared before acceptance for publication.

**Response**

We are thankful for the pertinent comments from Reviewer #1. Your suggestions are welcome additions to the manuscript and will help improve it significantly. Please find below a detailed response to your comments and how they impacted the manuscript.

**Major Comments**

**(Major Comment #1)**

First, the manuscript does not give a description of how the model initial condition as well as large scale forcing is constructed. The authors should describe how vertical wind shear is maintained during their simulation. Also, how is radiative heating computed? These are necessary information for reproducibility.

**(Response to Major Comment #1)**

We agree that the initial conditions should be clearly shown for reproducibility. We show the surface fluxes in Figure 1 in the manuscript and the initial conditions of the thermodynamic and wind profiles are shown in Figure 2. The radiation scheme is described in detail in Fu and Liou (1992) and Fu et al. (1997) as mentioned in Heus et al. (2010). This latter reference is the main reference for DALES and it describes all its components. New additions to the code that are relevant to this manuscript are then described in (de Bruine et al., 2019).  We added the references of Fu and Liou and Fu et al. to the text for clarity together with a few more comments about the model itself.

The vertical wind shear is maintained by the large scale forcing shown in Figure 2. The geostrophic wind conditions given in Figures 2c,d are maintained throughout the numerical experiments, except within the boundary layer that vary according the

turbulent conditions. Within the boundary layer, the winds tend to decelerate throughout the runs, but in the free troposphere the large-scale winds remain constant.

**(Major Comment #2)**

Second, the manuscript does not discuss/present general results of their simulation such as profiles of cloud water, cloud fraction, droplet number, fluxes, etc., time series of liquid water path, surface precipitation, etc. How can readers accept the new results without confirming reasonability of simulations? How are these LESs compared with observations?

**(Response to Major Comment #2)**

Thank you for this comment. Based on your suggestions, we have added two new figures to the manuscript, both of them located in the new Section 3.1. The first figure shows the time series of domain-averaged liquid water path (LWP), rainwater content at the surface ($RWC_{sfc}$) and the 99% percentile of cloud top height (CTH). The second figure shows vertical profiles of cloud liquid water content (LWC) and cloud fraction. Together with the simulated LWC profile of the NS run, we have added observations from the HALO aircraft during the GoAmazon2014/5 campaign (Martin et al., 2016; Wendisch et al., 2016). Hopefully those new figures help the readers have a better overall understanding of the simulations as well as how they compare with observations. For more details, please refer to the text in the new Section 3.1.

**(Major Comment #3)**

Third, their cloud tracking method discards clouds outside of the 10-km radius circle centered at the domain center. The authors justify this limitation due to the periodic boundary condition and clouds crossing the boundary. Since the domain is 21.6 km x 21.6 km, this means 33% of the domain is not used for their Lagrangian analysis. This leads to less or insufficient statistical sampling, which makes their results less significance. This also leads to statistical bias; mean of all identified clouds with the cloud tracking model has to be equal to the domain mean. The solid and dashed lines in Fig. 4 have to match up. There are ways to include clouds that cross the model lateral boundary. For instance, 1) tile 9 identical snapshots in a square, 2) classify all clouds over 9 tiles, 3) remove clouds whose part is not in the central tile, and 4) remove duplicated clouds that cross the lateral boundary (there are 2 identical clouds for a cloud crossing the boundary; at the corner there are 4 identical clouds).

**(Response to Major Comment #3)**

Thank you for the suggestion. We have updated the tracking method to better account for clouds crossing the boundaries. The ForTraCC algorithm used to do the tracking provides the specific horizontal pixels where the cloud is present. Based on this information, we now only exclude clouds that have any pixel touching the boundaries at

any point within their lifecycle. As seen in the manuscript, this has increased the number of clouds in more than 1000 per run. All manuscript figures have been updated after this change and the overall conclusions remain unchanged.

One side effect of this change is that it has further increased the contrast between the cloud tracking results and the domain-wide properties. This is because the added 1000+ clouds in each run are mostly smaller clouds. Therefore, the dashed and solid lines in the old Figure 4 (now Figure 6) are now further apart. Note that the larger and longer-lived the clouds are, the most likely they are at touching the boundaries. Therefore, it is natural that the tracking results will be biased towards the smaller and shorter-lived clouds. We think this is acceptable because we are indeed computing characteristics of most of the clouds present in the domain. Even though the few and large clouds do contribute a lot to the domain-wide characteristics, they do not contribute much to the quantifications we are interested in. Those quantifications are the change in cloud metric as function of VWS in an overall sense – i.e., how do the majority of clouds change with VWS? In the manuscript we argue that this baseline shallow cloud field is affected by VWS (even though with relatively small averaged changes), which then supports the formation of the deeper cloud. In that sense, it could interpreted as the deep cloud "feeding" off the shallow cloud field, growing in detriment of the shallower clouds. This would be a similar concept than rain droplets growing off of the smaller clouds droplets by collision-coalescence. Therefore, we can consider the deep cloud to be a singular entity and not really a part of the "shallow cloud field" we are interested in simulating.

This discussion has been updated in the text – please refer to the manuscript with tracked changes for full details (especially Section 3.2).

**(Major Comment #4)**

Fourth, as they discussed in the text, the model horizontal resolution may be too coarse for quantitative argument. For example, wind shear broadens the equivalent diameter up to 100 m on average, which is just 2 grid width.

These caveats have to be cleared before publication. The 4th caveat can be omitted by shifting to qualitative arguments.

I would recommend a major revision. However considering time required for upgrading their cloud tracking model and additional analysis, a longer period may be required.

**(Response to Major Comment #4)**

We agree that the averaged changes in cloud dimensions are small. This actually surprised us since we simulate a deeper cloud in the HS run by changing VWS alone. With respect to this, we would like to clarify the two aspects of the manuscript (as mentioned by the Reviewer in his/her general comment: the quantitative and qualitative aspects. The quantitative part focused on the overall shallow cloud field as discussed in our response #3 above. The more qualitative part discusses the effects that

the shallow cloud fields have on supporting the deeper clouds. We show that the HS run presents the deepest cloud in the simulations, being the only one producing significant surface precipitation. Note that the formation of this cloud is the result of a shallow cloud field that has been slightly affected by higher VWS. Our plausible explanation behind this feature is the quantification of total evaporation/condensation profiles which have a direct effect on the profiles of temperature and humidity. In short, this means that our manuscript deals with quantification until Figure 13 in the new manuscript and is more qualitative starting in the discussion of Figure 14.

We have updated Section 3.2 and the text in general to clarify this distinction to the reader.

While the averaged changes are equivalent to about the width of 2 pixels, it does not necessarily mean that the 50 m resolution is not enough to get some information. The quantifications would be better defined in 25 m, of course, but even in 50 m we can see a quantifiable influence of vertical wind shear on the cloud dimensions. This is because the statistical distribution of cloud dimension changes significantly between the runs, but the averaging produces values close to the model resolution. To illustrate, let us consider a hypothetical situation where we are comparing two groups of 10 pixels. Let us further consider that the hypothetical model is only capable of handling integer numbers. Say the two 10-pixel groups are defined like so: 1) 10 pixels containing only 1's; and 2) 5 pixels containing 1 and 5 pixels containing 2. It is reasonable to say that both groups are significantly different. However, the average value of group 1 is 1, while the average of group 2 is 1.5, i.e., only half of the model resolution. We think a similar aspect is present in our simulations, where the averaged changes are relatively small compared to the model resolution even though the statistics are visibly different. The fact that the metrics are mostly proportional to VWS, i.e., MS producing slightly larger clouds than NS and HS producing slightly larger clouds than MS, brings more consistency to the results as well.

**Minor Comments**

*Responses are italicized and in red.*

line 13: Remove "DALES". *Done.*

line 14: Remove "with" from "The resulting cloud field is analyzed with by applying...". *Done.*

line 27: Remove "a the" from "However, open questions still remain given that a the individually deepest clouds...". *Done.*

line 348: Add "9a" between "This figure" and "shows that the core represents between...". *Done.*

line 357: Change "Figure 9" to "Figure 9a". *Done.*

line 385: Change "continuous" to "contour". *Changed to "coloured shapes".*

line 446: Typo. Change "could" to "cloud". *Thank you.*

line 448: "it also tends to increase cloud clustering." If clustering is caused by cloud merging, this can be seen in the number of merger from the tracking data. *Agreed. We could probably see an effect in the number of mergers. The cloud clustering due to wind shear will be the subject of a new follow-up study, however.*

Figures: Yellow with a white background is hard to see. Change yellow to other color. *We opted for this color scale (called viridis) because it is better for readers with color blindness. We have decided to maintain the colouring as is in order to maintain a cohesive style throughout the manuscript.*

Figure 5: Plot time series of CAPE. *Thank you for the suggestion. But we think adding CAPE now wouldn't add much to the present discussion and could lead to convolution. We discuss CAPE only close to the end of the manuscript.*

Figure 11: The black contour is hard to see. Change its color. *Changed to green, now it's easier to read.*

**Reviewer #2**

**Overall comments**

This manuscript studies the evolution of shallow cumulus clouds over the Amazon in response to vertical wind shear. The role of wind shear is not well-studied and this study can make an important contribution to the field. The results indicate that a high value of vertical wind shear leads to (marginally) larger and shallower clouds, with a relatively smaller core area. They also find that there is more (localized) evaporation in the cloud layer under shear which may destabilize the cloud layer and have consequences for subsequent convection. Similar to what previous studies have found, the results also suggest there is not a linear relationship between cloudiness and wind shear, whereby high vertical wind shear can also increase the total water content present in the domain.

I think the authors did a nice job at explaining the cloud tracking algorithm that they use, at synthesizing the results and reflecting on the findings in light of the uncertainties in the simulation, the tracking and the complexity of the problem. Yet I agree that there is important information missing regarding the setup of the winds that requires clarification, which I outline below. There are also aspects of the sampling of core and margins that are unclear and I believe there are opportunities for using the cloud tracking statistics to explain how clouds of different sizes and LWC contribute to the overall increase in total water in the domain. I would advise a major revision based on the main comments below.

**Response**

We are thankful for the insightful comments from Reviewer #2. You present reasonable suggestions that have all been taken into account. This has led to a significant improvement in the manuscript. Please see below all the details about how we approached your suggestions.

**Major Comments**

**(Major Comment #1)**

The manuscript does not describe the evolution of wind profiles, the large-scale wind forcing and the surface stress. One aspect of the setup that is not clarified is that the winds will evolve during the course of the simulation: they will be mixed and slowed down throughout the BL in response to surface stress and the winds will gain an ageostrophic component. The shear present at noon would be different from the initial shear profiles.

The development of the wind profiles during the simulation influence how we interpret the differences in clouds that are presented as a result of differences in shear in the cloud layer.

A directional shear is prescribed in the initial profiles that turns counter-clockwise with height: usually you have a veering of the wind with height e.g. a clockwise wind turning (while friction would lead to a backing of the wind and a counter-clockwise wind turning towards the surface). Your geostrophic winds at 5 km imply that upon friction near the surface (or turbulence in the boundary layer) a westerly wind component will develop. Could this decrease the shear throughout the simulation period and lead to smaller shear differences as the simulation evolves?

Shear in the mixed layer can trigger indirect (as you mention, non-linear) effects. Helfer and Nuijens (2021) find that under forward shear in the subcloud layer there is the tendency for convection to become more organized and deepen more (although vertical updrafts within clouds are still hampered due to the slanting of clouds). The positioning of clouds relative to their sub-cloud layer roots can play a role in convective deepening by promoting stronger updrafts already in the subcloud layer. Under forward shear in the sub-cloud layer the updraft and downdraft region become more separated and the coherent circulations and resulting convergence may be strengthened.

https://agupubs.onlinelibrary.wiley.com/doi/full/10.1002/2015JD023253

The different wind speeds that develop at different heights is also relevant for the numerical diffusion of clouds, which is an issue with LES and the use of an all-or-nothing cloud scheme. The differences in cloud diameters between the shear cases is only 1 or two grid cells, something which I can imagine would be influenced by the differences in wind speed in the cloud layer (and such differences will be there even though a Galilean transform is applied, because the wind profiles are sheared).

**(Response to Major Comment #1)**

We have explained in Section 2.2 that the vertical wind profiles at the free troposphere remain unchanged throughout the simulations. But indeed, the winds decelerate within the boundary layer. The figure below shows the vertical wind speed (*wspd*) profiles in the first 2000 m for four different times in the day. For clarity, we only plot the NS, HS and HSR runs. The figure shows that the wind speed above the boundary layer remains constant throughout the day and there is a wind deceleration within the boundary layer. The runs with increased wind shear (HS and HSR in the figure) are characterized by an enhancement of 1 m/s extra wind speed in the boundary layer because of the entrainment of air masses characterized by a larger geostrophic wind. This wind enhancement leads to a more rapid and deep development of the convective boundary layer as mentioned in the manuscript, as well as the effects on convective organization mentioned by the Reviewer. We think this warrants a comment in the manuscript, but we have decided to not include another figure since the manuscript already has a lot of figures (14 after this revision). We have made comments on this topic in Section 2.2 – please check the tracked changes file for full details.

[Figure]

In terms of wind direction, we found that the westerly component only forms above 4500 m (see figure below), so we do not think it will have a significant effect on our simulations.

[Figure]

We are thankful for the very interesting references the Reviewer suggests. The Chen et al. (2015) study has analyzed the effect of VWS in the same 0-5 km layer as we use here and Helfer and Nuijens (2021) has found a very important relation between forward shear and cloud deepening. The latter reference, in particular, is relevant to our study, at least in the formation of the deepest cloud in the HS run. One caveat is that our clouds are non-precipitating, so there are no cold pool mechanisms. But, for the deepest cloud in HS, the separation between its roots and the precipitating downdrafts could play a role indeed. This is now mentioned in the discussion section with the reference. We decided to not cite the Chen et al. (2015) reference because of the different type of clouds simulated – their cloud type of interest are much deeper than the ones we are simulating here.

**(Major Comment #2)**

Can the authors clarify the core and margin selection and the results in Figure 8: am I right that the diameter of the core and margin sum up to be larger than the cloud diameter? This means that margins are occupying an area that is roughly the cloud diameter - core diameter (not really the margin or edge), or even larger, which would imply that there is overlap in the classification. It was my understanding that you try to contrast edges versus cores of the cloud, how should I be interpreting this?

**(Response to Major Comment #2)**

The cloud core and margins identification is described in Section 2.3. We define the cores as pixels containing positive vertical velocity, supersaturation above 0% and positive buoyancy. The margins identification is the opposite (negative in all three variables).

The Reviewer is right in that at some points the core + margins dimensions don't exactly sum up to the overall cloud dimension in the old Figure 8. The reason behind it is that the data sampling is slightly different for the cloud, core and margins. The restrictive nature of our cloud core/margin classification means that some pixels will not even have a classification. This can occur, for instance, close to the cloud outer edges where the buoyancy can be negative, but the vertical velocity is still slightly positive (Heus and Jonker, 2008). With different statistical samplings and independent calculations of the average dimensions of the cloud, cores and margins, there can be some differences. Additionally, we have reported values of equivalent diameter (i.e., diameter of a circle with the same area), so different shapes between the cloud, cores and margins could play a role as well. On the other hand, the figure shows that the sum of core + margins dimensions approximately sum up to the cloud dimension.

**(Major Comment #3)**

It would be nice if the authors make (more) use of tracking algorithm to delineate the contribution of clouds sizes/depths to LWC/LWP under different shear and to indicate whether there is more organization, which in studies of deep convection is clearly elated to the wind shear imposed. For instance, are there more merging events?

In section 3.2 you argue that whole-domain properties such as the total domain water are heavily influenced by infrequent and large clouds. I don't think you actually show his (yet). You could show this by plotting the cumulative cloud liquid water as a function of the cloud equivalent diameter (using all individual clouds). I'm not so sure whether it is just a single deep cloud that is responsible for the larger domain total water under high shear, because you also observe that under HS there are more frequent clouds with (intermediate) cloud depths of 600 - 1200 m: could they cumulatively explain a larger total water?

And can you show whether the larger domain water content under shear is contributed by the larger core dimensions in absolute term, or the larger margin dimensions in absolute terms?

In section 3.2 you also discuss that there is more evaporation under higher shear because clouds have a larger area due to tilting. At the bottom of page 13 you argue that there is more evaporation because of more liquid water content in the atmosphere (Figure 5). The cores are relatively small, but there is more cloud overall. Related to the above, it is not clear which clouds - and whether it is their cores or their margins - contribute most to the overall LWC in the domain. Can this be shown?

**(Response to Major Comment #3)**

We agree with the Reviewer in that we could make more use of the tracking algorithm. On the other hand, the cloud clustering and potentially more merger occurrences are outside the scope of this work. We are currently working on a follow-up manuscript where we will directly address the shallow cloud clustering. Therefore, we believe it is better to reserve this discussion for the next publication. In that way, we will be able to discuss this topic in more detail and depth. On the current manuscript, we have chosen to limit the analysis to the quantification of cloud/core/margins metrics as function of VWS and to the discussion about indirect VWS effects. On the follow-up manuscript we want to investigate other effects on cloud clustering aside from VWS too.

In the revised manuscript, the contribution of the deepest clouds to the high total domain water (TDW) values is more apparent – see the new Figure 6, for a somewhat indirect example. This figure shows the normalized histogram of LWP for all pixels in the domain versus only the pixels that are part of tracked clouds. This covers some of the Reviewer's comments about the contribution of large clouds to the LWP peaks. Note that the normalized counts for LWP > 400 g m$^{-2}$ are almost one order of magnitude higher in the continuous curves (all domain pixels) as compared to the dashed curves (tracked pixels) in the new Figure 6. As discussed in the responses to Reviewer #1, the tracking algorithm is biased towards smaller clouds because of the exclusion of clouds crossing the domain borders. Since larger clouds have a larger chance to cross the boundary domain (not only are they larger, but they are also longer lived), such clouds are prone to be excluded from the tracking analysis. Therefore, we can conclude that the highest-LWP pixels are associated to the infrequent deepest clouds.

Of course, this is only a partial response to the Reviewer's question, which is why we calculated the cumulative contributions of different cloud size classes to the TDW for the NS and HS runs. To do that calculation, we followed the steps: 1) we fist identify the time step with maximum TDW in the NS and HS runs (14:04 for NS and 14:42 for HS, in local time); 2) for that time, we define diameter classes that go from 100 m to 2000 m in 100 m intervals; 3) for each diameter class, we calculate the corresponding TDW contribution by summing all tracked clouds' total water within and below that class. This resulted in the figure below.

Firstly, we note that there was no tracked cloud larger than 1700 m in equivalent diameter for both runs at the maximum TDW time, which explains the flattening of the curves above such size. Secondly, there is a massive difference between the NS and HS runs. This is explained by the lack of a deep cloud in the NS run as compared to HS. In the new Figure 4 in the manuscript, we show that there is no precipitation in the NS run, therefore no particularly deep cloud. In that case, the contributions of relatively smaller clouds (the ones tracked by our algorithm) does indeed reach high values of almost 90%. However, when there is the presence of a deep cloud like in the HS run, this percentage falls to slightly above 10%. This confirms that the deepest cloud (and maybe a few more clouds between 1700 m in diameter and the size of the deepest cloud) in the HS is indeed responsible for most of the TDW. We think that the addition of the new Figure 4, together with the TDW time series figure is enough evidence to that fact for this manuscript.

[Figure]

Since the tracking algorithm did not follow the deepest cloud in HS, it is hard to compare the contributions from its core and margins without doing extra calculations. For the sake of time in this review, we think it is more appropriate to discuss it in qualitative

arguments. The most important aspect of the TDW value is the cloud volume itself. For instance, if for any reason there is a small cloud with high LWC values and a large cloud with lower LWC values, the latter will most likely have the higher TDW. Applying that concept to the deepest cloud in HS and taking into account that the margins are about 2x as large (horizontally) as the cores, it is possible that the margins have a larger contribution to the TDW. Again, we did not quantify this, but seems like a reasonable assumption for now.

**(Major Comment #4)**

With respect to the results that are in Figure 12: Figure 4 showed that under higher VWS there is a larger difference between all cloudy pixels and tracked pixels, also at larger LWP's. There must thus be a lot of not-considered cloud pixels in the simulation that also evaporate and contribute to the heating and moistening tendencies.

Have you compared the evaporation differences under shear to the domain-mean tendencies of humidity /temperature? All simulations have the same surface moisture/heat flux, but the humidity and temperature may be distributed differently. Do the differences in Fig 12 carry over to the total domain-mean tendencies or do the cloudy pixels not considered in the tracking also play a role?

The difference between localized (near cloud) changes in evaporation and domain-mean humidity and temperature tendencies is unclear in the results, but better discussed in the summary/ discussion.

**(Response to Major Comment #4)**

Thank you for the comment, this is a good point. But Figure 12 is calculated directly from the entire domain fields and not from the tracked clouds. Therefore, all clouds contribute to the curves shown.

**Minor Comments**

*Responses are italicized and in red below.*

1) A rigorous grammar & spelling check would be needed before publication. *Thank you. We have gone through the manuscript again and have improved the grammar and spelling to the best of our ability.*

2) Section 2.3: I would say that these days 21 x 21 km^2 is not that large. For shallow convection the typical domain size is now at least 50 going up to 125 km in one direction to allow for mesoscale dynamics to take place. *The Reviewer is correct, recent studies are using domain sizes larger than ours. We have rewritten to "given the relatively large number of clouds".*

3) P7L213: About the 10 km radius: At 7 m/s this corresponds to about 23 min, so that is the maximum length to be tracked .. Seems ok with respect to a 20 min life cycle of small clouds? (I guess you will present lifetime statistics later, but would fit here as well). *Given Reviewer #1's comments, we have changed the 10-km radius methodology. Now we only exclude clouds that at any point during their lifecycle had any pixel touching one of the boundaries. This has increased the number of clouds by more than 1000 per run and has also increased the maximum trackable lifetime. However, as discussed before, the largest clouds are still not tracked by the algorithm because they are more likely to touch the boundaries at some point.*

4) Section 2.3.1: L 224: "we exclude …" Do you repeat the process by excluding the larger shape from the 3D volume? (is that smaller shape tracked on its own and kept in the statistics despite being in close proximity to a larger clouds at some point?) I am asking because of the difference between isolated clouds and all tracked clouds in Figure 4. How much of that is because of excluding splitting or merging cells versus removing cells in close proximity or removing stacked clouds? I am asking because of the difference between isolated clouds and all tracked clouds in Figure 4. How much of that is because of excluding splitting or merging cells versus removing cells in close proximity or removing stacked clouds? *We have simplified this process. In the vertical, now we only exclude multilayered columns (only take the first cloud top of that column). In the horizontal, we set to 0 all variables in the pixels that are not identified by the tracking algorithm as a cloudy pixel. But as discussed in this document, this did not change the bias in the old Figure 4. This bias is mostly related to the longer duration and larger areas of the deepest clouds. Please refer to our response #3 to Reviewer #1 and to the tracked manuscript (specifically Section 3.2) for more details.*

5) Would section 3.1 be better placed in section 2 along with the description of the algorithm? *Section 3.1 is now Section 3.2 in the revised manuscript. Since it is already analyzing the outputs from the tracking algorithm, we have decided to to leave it in the results part instead of the methodology.*

6) Figure 8: It is confusing that the colours which were used before to indicate the amount of shear now denote the relative time duration. *We decided to use the same color scale (viridis) in all figures because it is better suited for colorblind readers. In some figures the colors represent different runs and in other it represents different times. Hopefully the figure descriptions are enough to clarify this distinction to the readers.*

7) I found that Figure 10 received quite some discussion but did not really aid the rest of the story. *Figure 10 is related to the story of this manuscript in that it further highlights the use of the tracking algorithm. It shows that the core dilution occurs throughout the entire cloud lifecycle with similar magnitudes. Because this figure is only possible with a tracking algorithm, we think it is valuable to keep it in the manuscript.*

8) Figure 11 is quite busy and complex. Could it be changed to a conditional PDF (conditioned on bins of LWC) that show the distributions of vertical velocity for the NS, MS, HS cases as just a line plot? Or some other version that more quickly allows one to

see the main result. BTW: does it seem that the differences in vertical velocity/updrafts are mainly pronounced at larger LWC? *The intent of the figure is to show the w and LWC distribution at the same time – i.e., 2D histograms. If we were to change it to line plots, we would have to draw several lines that would also be complex. Therefore, we think it's best to keep the figure as is. However, we did change the black curves to green, which helps in differentiating the different curves.*

9) P13 393: "VWS not only reduces the dimensions of the core": you miss denoting "relative" I think, because Figure 8b shows larger dimensions for the cores under shear. *Good point, thank you. We have updated it.*

10) P15 last (bottom) paragraph: What you write here could have also been addressed with "conventional" cloud (core) sampling in LES. Can you better emphasize the value of the tracking here? As mentioned, I think what can be really valuable is quantifying what are the cloud (sizes) that are most affected by shear: you show that while the small clouds are dissipated more quickly, the intermediate clouds may benefit and get larger. While clouds are overall shallower under more shear, there are a few much deeper clouds that develop under shear. *Thank you for the suggestion. We have updated this paragraph during the revision process. However, we decided to keep the paragraph more expansive to discuss our manuscript in the context of other publications. This is because the last paragraph is often used to give overall comments of the larger scope of the manuscript, which we think is the best option here.*

11) P16 512: "by changing the conditions of cloud formation": I think this can and should be explained a little further…. what conditions? *We changed it to "feedback mechanisms from cloud evaporation and BL formation". Such feedback mechanisms are discussed both before and after this sentence.*

**References mentioned in this document**

de Bruine, M., Krol, M., Vilà-Guerau de Arellano, J., and Röckmann, T.: Explicit aerosol–cloud interactions in the Dutch Atmospheric Large-Eddy Simulation model DALES4.1-M7, Geosci. Model Dev., 12, 5177–5196, https://doi.org/10.5194/gmd-12-5177-2019, 2019.

Fu, Q. and Liou, K. N.: On the Correlated k-Distribution Method for Radiative Transfer in Nonhomogeneous Atmospheres, Journal of Atmospheric Sciences, 49, 2139–2156, https://doi.org/10.1175/1520-0469(1992)049<2139:OTCDMF>2.0.CO;2, 1992.

Fu, Q., Liou, K. N., Cribb, M. C., Charlock, T. P., and Grossman, A.: Multiple Scattering Parameterization in Thermal Infrared Radiative Transfer, Journal of the Atmospheric Sciences, 54, 2799–2812, https://doi.org/10.1175/1520-0469(1997)054<2799:MSPITI>2.0.CO;2, 1997.

Heus, T. and Jonker, H. J. J.: Subsiding Shells around Shallow Cumulus Clouds, Journal of the Atmospheric Sciences, 65, 1003–1018, https://doi.org/10.1175/2007JAS2322.1, 2008.

Heus, T., van Heerwaarden, C. C., Jonker, H. J. J., Pier Siebesma, A., Axelsen, S., van den Dries, K., Geoffroy, O., Moene, A. F., Pino, D., de Roode, S. R., and Vilà-Guerau de Arellano, J.: Formulation of the Dutch Atmospheric Large-Eddy Simulation (DALES) and overview of its applications, Geosci. Model Dev., 3, 415–444, https://doi.org/10.5194/gmd-3-415-2010, 2010.

Martin, S. T., Artaxo, P., Machado, L. A. T., Manzi, A. O., Souza, R. A. F., Schumacher, C., Wang, J., Andreae, M. O., Barbosa, H. M. J., Fan, J., Fisch, G., Goldstein, A. H., Guenther, A., Jimenez, J. L., Pöschl, U., Silva Dias, M. A., Smith, J. N., and Wendisch, M.: Introduction: Observations and Modeling of the Green Ocean Amazon (GoAmazon2014/5), Atmos. Chem. Phys., 16, 4785–4797, https://doi.org/10.5194/acp-16-4785-2016, 2016.

Wendisch, M., Pöschl, U., Andreae, M. O., Machado, L. A. T., Albrecht, R., Schlager, H., Rosenfeld, D., Martin, S. T., Abdelmonem, A., Afchine, A., Araùjo, A. C., Artaxo, P., Aufmhoff, H., Barbosa, H. M. J., Borrmann, S., Braga, R., Buchholz, B., Cecchini, M. A., Costa, A., Curtius, J., Dollner, M., Dorf, M., Dreiling, V., Ebert, V., Ehrlich, A., Ewald, F., Fisch, G., Fix, A., Frank, F., Fütterer, D., Heckl, C., Heidelberg, F., Hüneke, T., Jäkel, E., Järvinen, E., Jurkat, T., Kanter, S., Kästner, U., Kenntner, M., Kesselmeier, J., Klimach, T., Knecht, M., Kohl, R., Kölling, T., Krämer, M., Krüger, M., Krisna, T. C., Lavric, J. V., Longo, K., Mahnke, C., Manzi, A. O., Mayer, B., Mertes, S., Minikin, A., Molleker, S., Münch, S., Nillius, B., Pfeilsticker, K., Pöhlker, C., Roiger, A., Rose, D., Rosenow, D., Sauer, D., Schnaiter, M., Schneider, J., Schulz, C., de Souza, R. A. F., Spanu, A., Stock, P., Vila, D., Voigt, C., Walser, A., Walter, D., Weigel, R., Weinzierl, B., Werner, F., Yamasoe, M. A., Ziereis, H., Zinner, T., and Zöger, M.: ACRIDICON–CHUVA Campaign: Studying Tropical Deep Convective Clouds and Precipitation over Amazonia Using the New German Research Aircraft HALO, Bulletin of the American Meteorological Society, 97, 1885–1908, https://doi.org/10.1175/BAMS-D-14-00255.1, 2016.